# ReTaSA: A Nonparametric Functional Estimation Approach for Addressing Continuous Target Shift

**Hwanwoo Kim**[1†]**, Xin Zhang**[2†]**, Jiwei Zhao**[3]**, Qinglong Tian**[4∗]

University of Chicago[1], Meta, Inc.[2], University of Wisconsin-Madison[3], University of Waterloo[4]

## Abstract

The presence of distribution shifts poses a significant challenge for deploying modern machine learning models in real-world applications. This work focuses on the target shift problem in a regression setting (Zhang et al., 2013; Nguyen et al., 2016). More specifically, the target variable $y$ (also known as the response variable), which is continuous, has different marginal distributions in the training source and testing domain, while the conditional distribution of features $\mathbf{x}$ given $y$ remains the same. While most literature focuses on classification tasks with finite target space, the regression problem has an *infinite dimensional* target space, which makes many of the existing methods inapplicable. In this work, we show that the continuous target shift problem can be addressed by estimating the importance weight function from an ill-posed integral equation. We propose a nonparametric regularized approach named *ReTaSA* to solve the ill-posed integral equation and provide theoretical justification for the estimated importance weight function. The effectiveness of the proposed method has been demonstrated with extensive numerical studies on synthetic and real-world datasets.

## 1 Introduction

Let $\mathbf{x} \in \mathbb{R}^p$ be the feature vector, $y \in \mathbb{R}$ be the target variable, and $p(\mathbf{x}, y)$ be their joint probability distribution. Classic supervised learning assumes the training and testing data are drawn from the same distribution (i.e., the joint distribution $p(\mathbf{x}, y)$ does not change across domains). However, in practice, distributional shifts between the training and testing data are pervasive due to various reasons, such as changes in the experimental objects, the data collection process, and the labeling process (Quinonero-Candela et al., 2008; Storkey, 2009). When the distributional shift occurs, the knowledge learned from the training source data may no longer be appropriate to be directly generalized to the testing data.

As a common type of distributional shift, the target shift assumes that the conditional distribution of features given the target variable $p(\mathbf{x}|y)$ remains the same in the training source and testing data domains, but the marginal distribution of the target variable $p(y)$ differs. Existing works (Tasche, 2017; Guo et al., 2020; Lipton et al., 2018; Du Plessis & Sugiyama, 2014; Iyer et al., 2014; Tian et al., 2023) on target shift primarily focused on the case where $y$ is categorical (also known as label shift or class prior change), while the cases with continuous $y$ (the regression task) have received less attention. However, the target shift problem in regression tasks is ubiquitous in real-world machine learning applications. For example, the Sequential Organ Failure Assessment (SOFA) score (Jones et al., 2009) is the continuous indicator associated with patients' wellness condition. Considering the scenario where the SOFA score is employed to determine the transfer destination of patients: those with higher SOFA scores, indicative of severe illness, are more likely to be transferred to large medical facilities with advanced medical equipment and highly skilled staff. Consequently, the distribution of SOFA scores among patients differs between large medical facilities and smaller clinics, while the distribution of features given the SOFA score remains the same across different hospitals. Nevertheless, the SOFA score predictive model trained on large medical facilities' datasets might lead to a bad performance if directly applied to smaller clinics' datasets. Another scenario for

---

∗Correspondence to qinglong.tian@uwaterloo.ca    † Equal Contribution

continuous target shift is the anti-causal setting (Schölkopf et al. 2012) where $y$ is the cause of $\mathbf{x}$. For example, it is reasonable to assume that height ($y$) is the cause of the bodyweight ($x$), and we can assume that $p(x|y)$ is domain invariant.

This work aims to fill this research gap and focus on the target shift problem in the regression tasks. We consider the *unsupervised domain adaptation* setting (Kouw & Loog, 2021), in which the training source dataset provides the feature-target (i.e., $(\mathbf{x}, y)$) data pairs while the testing dataset only has feature data (i.e., $\mathbf{x}$) available. The goal is to improve the predictive model's performance on the testing dataset by leveraging the knowledge gained from the training source data.

**Related Work**  The importance-weighted adaptation is one of the most popular approaches in addressing the target shift problem. However, most of the existing importance-weighted adaptation methods are designed specifically for dealing with the shift in the classification tasks (Saerens et al., 2002; Lipton et al., 2018; Azizzadenesheli et al., 2019; Alexandari et al., 2020; Garg et al., 2020). In the classification tasks, suppose the target variable $y$ has $k$ classes (e.g., $k = 2$ for binary classification). Then, the importance weight estimation can be simplified to an optimization problem with $k$ parameters. However, when the target variable $y$ is continuous (e.g., in a regression problem), these approaches are not feasible, as it remains unclear how to represent the continuous target space with a finite number of parameters. Existing work on the continuous target shift is scarce as opposed to its categorical counterpart. Except for some heuristic approaches, we found two works (Zhang et al., 2013; Nguyen et al., 2016) that addressed target shift using the idea of distribution matching (i.e., minimizing the distance between two distributions). However, neither of the two works provided sufficient theoretical justifications for their proposed methods: neither consistency nor error rate bounds have been established. Furthermore, both KMM (Zhang et al. 2013) and L2IWE (Nguyen et al. 2016) need to optimize a quadratically constrained quadratic program.

**Our Contributions**  In this paper, we propose the **Re**gularized Continuous **Ta**rget **S**hift **A**dapation (ReTaSA) method, a novel importance-weighted adaptation method for continuous target shift problem. The key component of the ReTaSA method is the estimation of the continuous importance weight function, which is crucial for shift adaptation. Using the estimated importance weight function, one can adjust the predictive model for the target domain via weighted empirical risk minimization (ERM). Our key results and their significance are summarized as follows:

1. We formulate the problem of estimating the continuous importance weight function as finding a solution to an integral equation. However, such an integral equation becomes ill-posed when the target variable is continuous. We introduce a nonparametric regularized method to overcome ill-posedness and obtain stable solutions.

2. We substantiate our method by theoretically examining the proposed importance weight function estimator. The theoretical findings, outlined in Theorem 1, demonstrate the statistical consistency of the estimated weight function. In addition to establishing consistency, we also quantify the convergence rate of our novel importance-weight function estimator, thereby enhancing the distinctive contributions of our research. To the best of our knowledge, this is the first work to offer a theoretical guarantee under the continuous case.

3. We develop an easy-to-use and optimization-free importance-weight estimation formula that only involves solving a linear system. Through comprehensive numerical experiments, we demonstrate that the ReTaSA method surpasses competitive approaches in estimation accuracy and computational efficiency.

## 2 THE TARGET SHIFT INTEGRAL EQUATION AND REGULARIZATION

We use subscripts $s$ and $t$ to represent the training source- and testing data domains, respectively. The target shift assumption says that $p_s(\mathbf{x}|y) = p_t(\mathbf{x}|y)$ while $p_s(y) \neq p_t(y)$, and the goal is to train a predictive model $\widehat{\mathbb{E}}_t(y|\mathbf{x})$ on the target domain. In terms of data samples, we have labeled data $\{(\mathbf{x}_i, y_i)\}_{i=1}^{n} \sim p_s(\mathbf{x}, y)$ from the source domain and unlabeled data $\{\mathbf{x}_i\}_{i=n+1}^{n+m} \sim p_t(\mathbf{x})$ from the testing data domain. Note that data from the testing data domain have no observation on the target variable $y$; thus, we cannot train $\widehat{\mathbb{E}}_t(y|\mathbf{x})$ directly and must leverage the feature-target data pair $(\mathbf{x}, y)$ from the training source domain.

We define the importance weight function as $\omega(y) \equiv p_t(y)/p_s(y)$, the target shift assumption implies that

$$\frac{p_t(\mathbf{x})}{p_s(\mathbf{x})} = \frac{\int p_t(\mathbf{x}, y)dy}{p_s(\mathbf{x})} = \frac{\int p_s(\mathbf{x}, y)\omega(y)dy}{p_s(\mathbf{x})} = \int p_s(y|\mathbf{x})\omega(y)dy. \tag{1}$$

Equation (1) has three terms: $p_t(\mathbf{x})/p_s(\mathbf{x})$, $p_s(y|\mathbf{x})$, and $\omega(y)$. Suppose $p_t(\mathbf{x})/p_s(\mathbf{x})$ and $p_s(y|\mathbf{x})$ are known, then (1) can be viewed as an integral equation to solve the unknown function $\omega(y)$. The importance weight function $\omega(y)$ is the key to target shift adaptation: letting $\ell(\cdot, \cdot)$ be the loss function, the risk of any predictive model $f(\cdot)$ on the testing data domain can be converted to a weighted risk on the training source domain: $\mathbb{E}_t[\ell\{f(\mathbf{x}), y\}] = \mathbb{E}_s[\omega(y)\ell\{f(\mathbf{x}), y\}]$, where $\mathbb{E}_s$ and $\mathbb{E}_t$ represent the expectation with respect to the training source and testing data distribution, respectively. Such equivalence implies that we can learn a predictive model for the target domain via weighted ERM using data from the source domain with the importance weight function $\omega(y)$. Most existing work deals with the classification tasks with finite $k$ classes, in which $y$ comes from a finite discrete space $\{1, \cdots, k\}$. Thus, the right-hand-side of (1) can be represented in a summation form $\sum_{y=1}^{k} p_s(y|\mathbf{x})\omega(y)$ and $\{\omega(y), y = 1, \cdots, k\}$ can be estimated by solving the $k$-dimensional linear equations (Lipton et al., 2018). However, when $y$ is continuous, the importance weight function cannot be reduced to a finite number of parameters as $y$ can take an infinite number of values. Such a difference pinpoints the difficulties that arise when $y$ is continuous.

**Operator Notations**   We first introduce a few operator notations for our problem formulation. Considering the training source distribution $p_s(\mathbf{x}, y)$, we use $L^2(\mathbf{x}, y)$ to denote the Hilbert space for mean-zero square-integrable functions of $(\mathbf{x}, y)$. The inner product for any $h_1, h_2 \in L^2(\mathbf{x}, y)$ is defined by $\langle h_1, h_2 \rangle = \int h_1(\mathbf{x}, y)h_2(\mathbf{x}, y)p_s(\mathbf{x}, y)d\mathbf{x}dy = \mathbb{E}_s\{h_1(\mathbf{x}, y)h_2(\mathbf{x}, y)\}$, and the norm is induced as $\|h\| = \langle h, h \rangle^{1/2}$. Similarly, we use $L^2(\mathbf{x})$ or $L^2(y)$ to respectively denote the subspace that contains functions of $\mathbf{x}$ or $y$ only. In this case, the inner product $h_1, h_2 \in L^2(\mathbf{x})$ or $L^2(y)$ is given by $\langle h_1, h_2 \rangle = \int h_1(\mathbf{x})h_2(\mathbf{x})p_s(\mathbf{x})d\mathbf{x}$ or $\langle h_1, h_2 \rangle = \int h_1(y)h_2(y)p_s(y)dy$. Additionally, we define two conditional expectation operators

$$T : L^2(y) \rightarrow L^2(\mathbf{x}), s.t. \, T\varphi = \mathbb{E}_s\{\varphi(y)|\mathbf{x}\}, \tag{2}$$

$$T^* : L^2(\mathbf{x}) \rightarrow L^2(y), s.t. \, T^*\psi = \mathbb{E}_s\{\psi(\mathbf{x})|y\}. \tag{3}$$

$T$ and $T^*$ are adjoint operators because

$$\langle T\varphi, \psi \rangle = \int \psi(\mathbf{x})\mathbb{E}_s\{\varphi(y)|\mathbf{x}\}\, p_s(\mathbf{x})d\mathbf{x} = \iint \varphi(y)\psi(\mathbf{x})p_s(\mathbf{x}, y)dyd\mathbf{x}$$

$$= \int \varphi(y)\mathbb{E}_s\{\psi(\mathbf{x})|y\}\, p_s(y)dy = \langle \varphi, T^*\psi \rangle$$

for any $\varphi \in L^2(y)$ and $\psi \in L^2(\mathbf{x})$. Using the operator notations, we rewrite (1) as

$$T\rho_0 = \eta, \tag{4}$$

where $\eta(\mathbf{x}) \equiv p_t(\mathbf{x})/p_s(\mathbf{x}) - 1$ and $\rho_0(y) \equiv \omega(y) - 1$ is the unknown function to be solved. We define $\eta(\mathbf{x})$ in such way to satisfy the mean-zero condition $\mathbb{E}_s\{\eta(\mathbf{x})\} = 0$ so that $\eta(\mathbf{x}) \in L^2(\mathbf{x})$.

**Identifiability**   With the above reformulation, a natural question is whether $\rho_0$ is identifiable, i.e., the solution of the equation (4) is unique. To further ensure uniqueness of the solution to (4), let the null set of $T$ be $\mathcal{N}(T) = \{\varphi : T\varphi = 0, \varphi \in L^2(y)\}$. Then $\rho_0$ is a unique solution to (4) if and only if $\mathcal{N}(T) = \{0\}$. Because $T$ is a conditional expectation operator, we can equivalently express the identifiability condition with the following *completeness* condition. We refer the readers to Newey & Powell (2003) for a detailed discussion on the completeness condition.

**Condition 1** (Identifiability Condition). *The importance weight function $\omega(y)$ is identifiable in (1) if and only if $\mathbb{E}_s\{\varphi(y)|\mathbf{x}\} = 0$ almost surely implies that $\varphi(y) = 0$ almost surely for any $\varphi \in L^2(y)$.*

**Ill-posedness & Regularization**   Even though the integral equation in (4) has a unique solution under Condition 1, we are still not readily able to solve for $\rho_0$. This is because (4) belongs to the class of Fredholm integral equations of the first kind (Kress, 1999), which is known to be ill-posed. More specifically, it can be shown that even a small perturbation in $\eta$ can result in a large change in the solution $\rho_0$ in (4). This phenomenon is referred to as an "explosive" behavior (Darolles et al., 2011),

for which we provide details in Section A the supplementary materials. Such sensitive dependence upon input (i.e., $\eta$) is problematic because, in practice, the true function $\eta$ is unknown, and an estimated version $\widehat{\eta}$ would be plugged in. No matter how accurate the estimation is, the estimated $\widehat{\eta}$ will inevitably differ from $\eta$, thus leading to a large error on $\rho_0$.

To address the ill-posedness of Equation (4), we adopt the Tikhonov regularization technique, which has been broadly applied in various tasks, for example, the ridge regression (Hoerl & Kennard, 1970), and the instrumental regression (Carrasco et al., 2007; Fève & Florens, 2010; Darolles et al., 2011). The Tikhonov regularization yields a regularized estimator $\rho_\alpha$ by minimizing a penalized criterion

$$\rho_\alpha = \underset{\rho \in L^2(y)}{\operatorname{argmin}} \|T\rho - \eta\|^2 + \alpha \|\rho\|^2,$$

where $\alpha$ is a regularization parameter. This minimization leads to the first-order condition for a unique solution to $(\alpha I + T^*T)\,\rho = T^*\eta$, where I is the identity operator. Furthermore, by the definition of $T$ and $T^*$ in (2)-(3), we can rewrite the first-order condition as

$$\alpha \rho(y) + \mathbb{E}_s\left[\mathbb{E}_s\left\{\rho(y)|\mathbf{x}\right\}|y\right] = \mathbb{E}_s\left\{\eta(\mathbf{x})|y\right\}. \tag{5}$$

## 3 ESTIMATION OF THE IMPORTANCE WEIGHT FUNCTION

This section focuses on how to estimate the solution function $\rho_\alpha$ using (5). Estimating $\rho_\alpha$ requires two steps: the first step is to replace (5) with a sample-based version; the second step is to solve for the estimate $\widehat{\rho}_\alpha$ from the sample-based equation. We employ the kernel method to find the sample-based version of (5). To this end, without loss of generality, we assume $\mathbf{x}$ and $y$ take values in $[0,1]^p$ and $[0,1]$ respectively. In particular, we use a generalized kernel function, whose definition is given below.

**Definition 1.** *Let $h$ be a bandwidth, a bivariate function $K_h : [0,1] \times [0,1] \to \mathbb{R}_{\geq 0}$ satisfying:*

$$1)\ K_h(x, \tilde{x}) = 0, \text{if } x > \tilde{x} \text{ or } x < \tilde{x} - 1, \text{ for all } \tilde{x} \in [0,1];$$

$$2)\ h^{-(q+1)} \int_{y-1}^{y} x^q K_h(x, \tilde{x}) dx = \begin{cases} 1 & \text{for } q = 0, \\ 0 & \text{for } 1 \leq q \leq \ell - 1, \end{cases}$$

*is referred as a univariate generalized kernel function of order $\ell$.*

Based on the above definition, the density estimates are given by

$$\widehat{p}_s(\mathbf{x}, y) = (nh^{p+1})^{-1} \sum_{i=1}^{n} K_{\mathbf{X},h}(\mathbf{x} - \mathbf{x}_i, \mathbf{x}) K_{Y,h}(y - y_i, y), \quad \widehat{p}_s(y) = (nh)^{-1} \sum_{i=1}^{n} K_{Y,h}(y - y_i, y),$$

$$\widehat{p}_t(\mathbf{x}) = (nh^p)^{-1} \sum_{i=n+1}^{n+m} K_{\mathbf{X},h}(\mathbf{x} - \mathbf{x}_i, \mathbf{x}), \quad \widehat{p}_s(\mathbf{x}) = (nh^p)^{-1} \sum_{i=1}^{n} K_{\mathbf{X},h}(\mathbf{x} - \mathbf{x}_i, \mathbf{x}),$$

where $p$ is the dimension of $\mathbf{x}$, $h$ is the bandwidth parameter, $K_{Y,h}$ is a univariate generalized kernel function, and $K_{\mathbf{X},h}$ is a product of univariate generalized kernel functions defined for each component of $\mathbf{X}$. In addition, we estimate $\eta(\mathbf{x})$ with $\widehat{\eta}(\mathbf{x}) = \widehat{p}_t(\mathbf{x})/\widehat{p}_s(\mathbf{x}) - 1$. Instead of simply calculating the ratio of two density estimates, there are other methods for computing the density ratio (e.g., Sugiyama et al. 2012). Due to the space limit, we focus on the simple approach $\widehat{\eta}(\mathbf{x}) = \widehat{p}_t(\mathbf{x})/\widehat{p}_s(\mathbf{x})$ and leave more details in Section C.2.2 of the supplementary materials.

Let $\rho$ be an arbitrary function in $L^2(y)$. Using the kernel density estimates, the Nadaraya–Watson estimates of $\mathbb{E}_s\{\rho(y)|\mathbf{x}\}$, $\mathbb{E}_s\{\widehat{\eta}(\mathbf{x})|y\}$, and $\mathbb{E}_s[\mathbb{E}_s\{\rho(y)|\mathbf{x}\}|y]$ are, respectively, given by

$$\widehat{\mathbb{E}}_s\{\rho(y)|\mathbf{x}\} = \frac{\sum_{i=1}^{n} \rho(y_i) K_{\mathbf{X},h}(\mathbf{x} - \mathbf{x}_i, \mathbf{x})}{\sum_{j=1}^{n} K_{\mathbf{X},h}(\mathbf{x} - \mathbf{x}_j, \mathbf{x})}, \quad \widehat{\mathbb{E}}_s\{\widehat{\eta}(\mathbf{x})|y\} = \frac{\sum_{i=1}^{n} \widehat{\eta}(\mathbf{x}_i) K_{Y,h}(y - y_i, y)}{\sum_{j=1}^{n} K_{Y,h}(y - y_j, y)},$$

and

$$\widehat{\mathbb{E}}_s[\widehat{\mathbb{E}}_s\{\rho(y)|\mathbf{x}\}|y] = \left\{\sum_{k=1}^{n} K_{Y,h}(y - y_k, y)\right\}^{-1} \left\{\sum_{\ell=1}^{n} \frac{\sum_{i=1}^{n} \rho(y_i) K_{\mathbf{X},h}(\mathbf{x}_\ell - \mathbf{x}_i, \mathbf{x}_\ell)}{\sum_{j=1}^{n} K_{\mathbf{X},h}(\mathbf{x}_\ell - \mathbf{x}_j, \mathbf{x}_\ell)} K_{Y,h}(y - y_\ell, y)\right\}.$$

Using the estimates above, we can obtain the sample-based version of (5), and the next step is to solve for the function $\rho$. By letting $y = y_1, \ldots, y_n$, we have the following linear system:

$$\alpha \left[\rho(y_1) \ldots \rho(y_n)\right]^{\mathrm{T}} + \mathbf{C}_{\mathbf{x}|y}\mathbf{C}_{y|\mathbf{x}} \left[\rho(y_1) \ldots \rho(y_n)\right]^{\mathrm{T}} = \mathbf{C}_{\mathbf{x}|y} \left[\widehat{\eta}(\mathbf{x}_1) \ldots \widehat{\eta}(\mathbf{x}_n)\right]^{\mathrm{T}} \qquad (6)$$

where $\mathbf{C}_{\mathbf{x}|y}$ and $\mathbf{C}_{y|\mathbf{x}}$ are both $n \times n$ matrices and their $(i, j)$th entries are given by

$$\left(\mathbf{C}_{\mathbf{x}|y}\right)_{ij} = \frac{K_{Y,h}(y_i - y_j, y_i)}{\sum_{\ell=1}^n K_{Y,h}(y_i - y_\ell, y_i)} \text{ and } \left(\mathbf{C}_{y|\mathbf{x}}\right)_{ij} = \frac{K_{\mathbf{X},h}(\mathbf{x}_i - \mathbf{x}_j, \mathbf{x}_i)}{\sum_{\ell=1}^n K_{\mathbf{X},h}(\mathbf{x}_i - \mathbf{x}_\ell, \mathbf{x}_i)}.$$

We can obtain the estimate of $\rho_\alpha$ on $\{y_1, \ldots, y_n\}$ by solving the linear system (6) as

$$\left[\widehat{\rho}_\alpha(y_1), \ldots, \widehat{\rho}_\alpha(y_n)\right]^{\mathrm{T}} = \left(\alpha\mathbf{I} + \mathbf{C}_{\mathbf{x}|y}\mathbf{C}_{y|\mathbf{x}}\right)^{-1} \mathbf{C}_{\mathbf{x}|y} \left[\widehat{\eta}(\mathbf{x}_1), \ldots, \widehat{\eta}(\mathbf{x}_n)\right]^{\mathrm{T}}. \qquad (7)$$

Note that the estimates on $\{y_1, \ldots, y_n\}$ in (7) are all we need because the importance weights are $\widehat{\omega}(y_i) = \widehat{\rho}_\alpha(y_i) + 1$, $i = 1, \ldots, n$; thus, for a family of regression models, denoted by $\mathcal{F}$, the trained regression model for the target domain can be obtained from the weighted ERM

$$\widehat{\mathbb{E}}_t(Y|\mathbf{X} = \mathbf{x}) = \operatorname*{argmin}_{f \in \mathcal{F}} \frac{1}{n} \sum_{i=1}^n \widehat{\omega}(y_i) \left\{f(\mathbf{x}_i) - y_i\right\}^2.$$

## 4 Theoretical Analysis

In this section, we provide a theoretical understanding of the proposed kernel-based importance weight function-based adaptation. In particular, we establish the consistency of $\hat{\rho}_\alpha$, a solution of the sample-based approximation, to the solution $\rho_0$ of the equation (4) under suitable assumptions. We define two estimated conditional expectation operators as

$$\widehat{T} : L^2(y) \to L^2(\mathbf{x}), s.t. \; \widehat{T}\varphi = \widehat{\mathbb{E}}_s \left\{\varphi(y)|\mathbf{x}\right\} \equiv \int \varphi(y)\widehat{p}_s(y|\mathbf{x})dy, \qquad (8)$$

$$\widehat{T}^* : L^2(\mathbf{x}) \to L^2(y), s.t. \; \widehat{T}^*\psi = \widehat{\mathbb{E}}_s \left\{\psi(\mathbf{x})|y\right\} \equiv \int \psi(\mathbf{x})\widehat{p}_s(\mathbf{x}|y)d\mathbf{x} \qquad (9)$$

Furthermore, by plugging $\widehat{\eta}(\mathbf{x}) = \widehat{p}_t(\mathbf{x})/\widehat{p}_s(\mathbf{x}) - 1$, we have the kernel-based approximation of equation (5) as

$$\left(\alpha\mathrm{I} + \widehat{T}^*\widehat{T}\right)\rho = \widehat{T}^*\widehat{\eta} \quad \text{or} \quad \alpha\rho(y) + \widehat{\mathbb{E}}_s \left[\widehat{\mathbb{E}}_s \left\{\rho(y)|\mathbf{x}\right\}|y\right] = \widehat{\mathbb{E}}_s \left\{\widehat{\eta}(\mathbf{x})|y\right\}. \qquad (10)$$

Now, we provide a theoretical justification for our proposed kernel-based regularization framework. Specifically, we first prove the consistency of the solution obtained from (10), denoted by $\widehat{\rho}_\alpha \equiv \left(\alpha\mathrm{I} + \widehat{T}^*\widehat{T}\right)^{-1} \widehat{T}^*\widehat{\eta}$. Toward this end, we state several necessary assumptions.

**Assumption 1.** *The joint source density $p_s(\mathbf{x}, y)$ and marginal source densities $p_s(\mathbf{x})$ and $p_s(y)$ satisfy $\int \int \left\{\frac{p_s(\mathbf{x},y)}{p_s(\mathbf{x})p_s(y)}\right\}^2 p_s(\mathbf{x})p_s(y)d\mathbf{x}dy < \infty$.*

**Remark 1.** *The implication of Assumption 1 is that two operators $T$ and $T^*$ defined in (2) and (3) are Hilbert-Schmidt operators and therefore compact, which admits the singular value decomposition.*

To establish consistency, we restrict the function space in which the solution $\rho_0$ resides. Consequently, we introduce the $\beta$-regularity space of the operator $T$ for some $\beta > 0$,

**Definition 2.** *For $\beta > 0$, the $\beta$-regularity space of the compact operator $T$ is defined as $\Phi_\beta = \left\{\rho \in L^2(y) \text{ such that } \sum_{i=1}^\infty \langle\rho, \varphi_i\rangle^2/\lambda_i^{2\beta} < \infty\right\}$, where $\{\lambda_i\}_{i=1}^\infty$ are the decreasing sequence of nonzero singular values of $T$ and $\{\varphi_i\}_{i=1}^\infty$ is a set of orthonormal sequence in $L^2(y)$. In other words, for some orthonormal sequence $\{\phi_i\}_{i=1}^\infty \subset L^2(\mathbf{x})$, $T\phi_i = \lambda_i\varphi_i$ holds for all $i \in \mathbb{N}$.*

**Assumption 2.** *There exists a $\beta > 0$ such that the true solution $\rho_0 \in \Phi_\beta$.*

**Remark 2.** *In general, for any $\beta_1 \geq \beta_2$, $\Phi_{\beta_1} \subseteq \Phi_{\beta_2}$ (Carrasco et al., 2007). Therefore, the regularity level $\beta$ governs the complexity of the space. A detailed discussion of the $\beta$-regularity space is given in Section A.1 of the supplementary materials.*

**Assumption 3.** *We assume* $\mathbf{x} \in [0,1]^p$, $y \in [0,1]$; *and the* $k$-*times continuously differentiable densities* $p_s(\mathbf{x}, y)$, $p_s(\mathbf{x})$ *and* $p_t(\mathbf{x})$ *are all bounded away from zero on their supports, i.e.,* $p_s(\mathbf{x}, y), p_s(\mathbf{x}), p_t(\mathbf{x}) \in [\epsilon, \infty)$, *for some* $\epsilon > 0$.

**Remark 3.** *The equation* (1) *is problematic when* $p_s(\mathbf{x})$ *goes to zero. Therefore we need the assumption that* $p_s(\mathbf{x})$ *is bounded from below by some constant* $\epsilon > 0$. *Also, we need the support of* $p_t(\mathbf{x})$ *to be a subset of the support of* $p_s(\mathbf{x})$. *When* $p_s(\mathbf{x})$ *goes to zere, while* $p_t(\mathbf{x})$ *is not zero, it corresponds to the out-of-distribution (OOD) setting (Zhou et al., 2023).*

As we adopt the class of multivariate generalized kernel functions formed by a product of univariate generalized kernel functions, we impose the following common assumptions which can be found in (Darolles et al., 2011; Hall & Horowitz, 2005)

**Assumption 4.** *With a bandwidth parameter* $h$, *a generalized kernel function* $K_h : [0,1] \times [0,1] \to \mathbb{R}_{\geq 0}$ *satisfies the following conditions:*

1. $K_h(x, y) = 0$ *if* $x \notin [y - 1, y] \cap \mathcal{C}$, *where* $\mathcal{C}$ *is a compact interval independent of* $y$.

2. $\sup\limits_{x \in \mathcal{C}, y \in [0,1]} |K_h(x, y)| < \infty$.

**Remark 4.** *Note that the domain restriction is primarily for the purpose of theoretical analysis. In fact, any compact domain would suffice. The continuous differentiability and boundedness assumptions on the density are to obtain uniform convergence of KDE estimators, which serves as an important tool to establish consistency. Such uniform convergence results based on the ordinary kernel functions need additional restrictions on the domain, while the generalized kernel functions can obtain uniform convergence over the entire compact support Darolles et al. (2011).*

To guarantee convergence in probability, we make assumptions on the decaying rate of the bandwidth parameter $h$ and regularization parameter $\alpha$ with respect to the sample size $n$.

**Assumption 5.** *The kernel bandwidth* $h$, *regularization parameter* $\alpha$ *and the source sample size* $n$ *satisfy* $\lim\limits_{\substack{h \to 0 \\ n \to \infty}} \log(n)/(nh^{p+1}) = 0$. *Furthermore, the target sample size* $m$ *satisfies* $\lim\limits_{\substack{h \to 0 \\ m \to \infty}} \log(m)/(mh^p) = 0$.

**Remark 5.** *The assumption implies that the growth rate of* $n$ *is faster than the decaying rate of* $h^{p+1}$ *so that* $nh^{p+1}$ *grows faster than the* $\log(n)$ *rate. Similar to* $n$, *we require the growth rate of* $m$ *to be faster than the decaying rate of* $h^p$. *As the assumption indicates, the price one must pay to establish a convergence rate in the high-dimensional setting is non-negligible.*

We formally state the consistency result, of which the proof is provided in Section B of the supplementary materials.

**Theorem 1.** *For* $\beta \geq 2$, *under Assumptions 1 - 5, one can show that* $||\widehat{\rho}_\alpha - \rho_0||^2$ *is of order*

$$
O_p\left( \alpha^2 + \left( \frac{1}{nh^{p+1}} + h^{2\gamma} \right) \left( \frac{1}{nh^{p+1}} + h^{2\gamma} + 1 \right) \alpha^{(\beta-2)} \right.
$$

$$
\left. + \frac{1}{\alpha^2} \left( \frac{1}{nh^{p+1}} + h^{2\gamma} + 1 \right) \left( \frac{1}{n} + h^{2\gamma} + \sqrt{\frac{\log m}{mh^p}} \left( h^\gamma + \sqrt{\frac{\log m}{mh^p}} \right) \right) \right). \tag{11}
$$

*In particular, if* $\max \left( h^{2\gamma}, \sqrt{\frac{\log(m)}{mh^p}} h^\gamma, \frac{\log(m)}{mh^p} \right) = o_p(\alpha^2)$ *and* $\lim\limits_{\substack{\alpha \to 0 \\ n \to \infty}} n\alpha^2 = \infty$, $\widehat{\rho}_\alpha$ *converges in probability to* $\rho_0$, *i.e.,* $||\widehat{\rho}_\alpha - \rho_0||^2 = o_p(1)$, *as* $\alpha, h \to 0$, $n \to \infty$, *and* $m \to \infty$.

**Remark 6.** *In the proof, we conduct the theoretical analysis by decomposing the error between* $\rho_0$ *and* $\widehat{\rho}_\alpha$ *into three parts. Each term, inside the* $O_p(\cdot)$ *expression in* (11), *indicates the errors manifested by each of the three parts. The first term inside the* $O_p(\cdot)$ *represents the error between the unregularized solution* $\rho_0$ *and the regularized solution* $\rho_\alpha$. *The second term appears due to kernel-based approximation of conditional expectation operators. The last factor reflects the cost of estimating conditional expectation operators and target/source densities. Naturally, the target sample size only gets involved in the last term.*

**Remark 7.** *To establish consistency, $n$ must grow faster than $\alpha^2$ to make $n\alpha^2$ divergent. In the meantime, the decay rate of regularization parameter $\alpha^2$ should be slower than all three terms $h^{2\gamma}, \sqrt{\log(m)/(mh^p)}h^\gamma$ and $\log(m)/(mh^p)$. Therefore, the assumption indicates that the regularization parameter should decay slower as the dimension of the feature becomes larger, i.e., a larger regularization effect is needed for a higher dimension. Furthermore, the assumption indicates that one can use a smaller regularization parameter for smoother densities with a larger order of the generalized kernel function.*

**Remark 8.** *Theorem 1 provides a bound for the estimation error's weighted $L^2$ norm (i.e., $\|\widehat{\rho}_\alpha - \rho_0\|^2 = \int \{\widehat{\rho}_\alpha(y) - \rho_0(y)\}^2 p_s(y) dy$). Compared with a function's ordinary $L^2$ norm (i.e. $\int \{\widehat{\rho}_\alpha(y) - \rho_0(y)\}^2 dy$), the weighted $L^2$ norm directly bounds the target domain generalization error. More details are given in Section B.1 of the supplementary materials.*

## 5 EXPERIMENTS

In this section, we present the results of numerical experiments that illustrate the performance of our proposed method in addressing the continuous target shift problem. We will first study with the synthetic data and then apply the methods to two real-world regression problems. Our experiments are conducted on a Mac-Book Pro equipped with a 2.9 GHz Dual-Core Intel Core I5 processor and 8GB of memory. We relegated the experimental details (e.g., hyperparameter tuning) and extensive experiments with high-dimensional datasets to Section C in the supplementary material.

### 5.1 NONLINEAR REGRESSION WITH SYNTHETIC DATA

We first conduct a synthetic data analysis with a nonlinear regression problem. Inspired by the setting in Zhang et al. (2013), we generate the dataset with $x = y + 3 \times \tanh(y) + \epsilon$, where $\epsilon \sim \text{Normal}(0, 1)$. The target $y_s$ for source data is from $\text{Normal}(0, 2)$, while the testing data has the target $y_t$ from $\text{Normal}(\mu_t, 0.5)$. We use $\mu_t$ to adjust the target shift between the training source and testing data. We set the size of the target testing data as $0.8$ of this study's training source data size, i.e., $m = 0.8 \times n$ unless otherwise specified. Figure 1 shows an example with $\mu_t = 0.5$ and $n = 200$. As for the target prediction, we adopt the polynomial regression model with degrees as $5$. In this simulation study, we consider three existing adaptation strategies as baseline methods: 1). Non-Adaptation, in which the source data trained model is directly applied for the prediction. 2). Oracle-Adaptation, of which the importance weight function is the ground truth. 3). KMM-Adaptation proposed by Zhang et al. (2013). 4). L2IWE-Adaptation proposed by Nguyen et al. (2016). It is worth noting that Oracle-Adaptation is infeasible in practice as the shift mechanism is unknown.

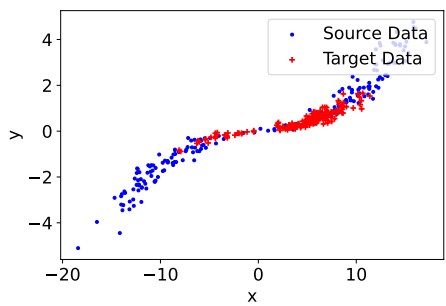

Figure 1: An illustration of the nonlinear regression synthetic data with $n = 200$ and $\mu_t = 0.5$. The blue dots are the source data, and the red plus marks are the target data.

**Evaluation Metrics** Two metrics are used to evaluate the adaptation performance in our experiments: 1). ***Weight MSE***: the mean square error of the estimated adaptation weights v.s. the weights from Oracle-Adaptation, i.e., $\sum_{i=1}^n \|\hat{\omega}(y_i) - \omega(y_i)\|^2/n$. Non-Adaptation calculates the weight MSE by assuming no shift as $\hat{\omega}(y) \equiv 1$. 2). ***Delta Accuracy*** ($\Delta$ Accuracy): the percentage of the improved prediction MSE compared with that of Non-Adaptation, and the larger value represents the better improvement. We conducted all experiments with 50 replications.

**Experimental Results** Our first study focuses on the adaptation methods' performances under different data sizes. We fix $\mu_t = 0.5$ and change the data size $n$ from $100$ to $1000$. The results are shown in Figure 2 (a)&(d). Our proposed method performs significantly better than L2IWE-Adaptation and KMM-Adaptation method in terms of both weight estimation and prediction accuracy. Compared with Non-Adaptation, KMM and L2IWE methods reduce the weight MSE by about $25\%$

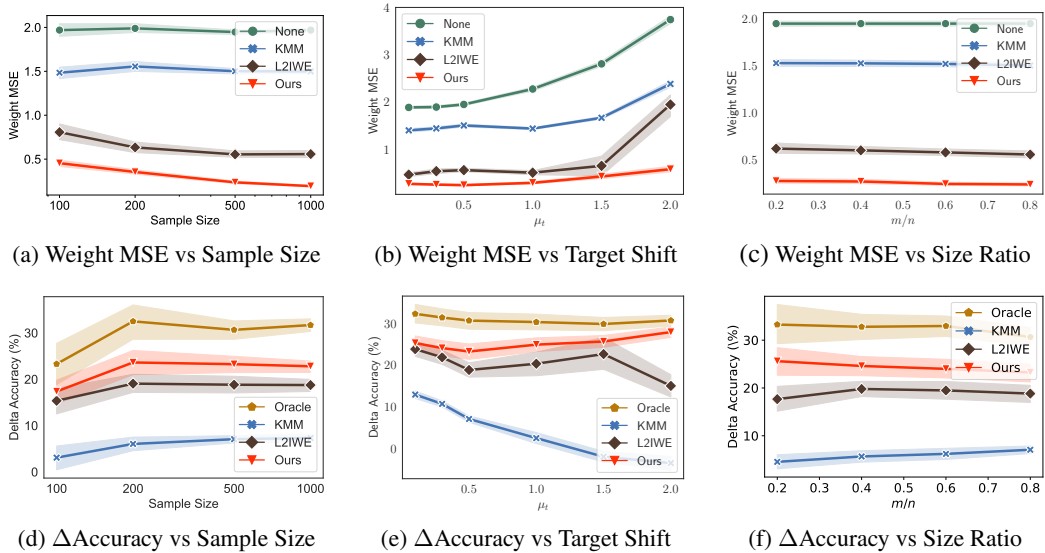

(a) Weight MSE vs Sample Size     (b) Weight MSE vs Target Shift     (c) Weight MSE vs Size Ratio

(d) ΔAccuracy vs Sample Size     (e) ΔAccuracy vs Target Shift     (f) ΔAccuracy vs Size Ratio

Figure 2: Performances comparison over difference adaptation methods. The first row uses weight MSE as a metric, and the second one shows the performance measuring by ΔAccuracy. (a)&(d) are the performance over different sample sizes. The sample sizes range from {100, 200, 500, 1000}. (b)&(e) are the performances over different target shifts $\mu_t$. The values of $\mu_t$ range from 0.1 to 2. (c)&(f) are the performances over different target-source data size ratio $m/n$ and the values range from 0.2 to 0.8. The solid curves are the mean values and the shadow regions are 95% CI error bands.

and 60%, respectively, while our method reduces it by more than 75%. For prediction accuracy, increasing data size from 100 to 1000, our method improves accuracy from about 15% to 22% while KMM-Adaptation improves less than 10%. The L2IWE method is better than the KMM method but still not as good as the proposed method.

Furthermore, we study the performances under different target shifts with the results shown in Figure 2 (b)&(e). Consistently, our proposed adaptation method still outperforms KMM and L2IWE. It can be seen from Figure 2 (b) that as $\mu_t$ increases, the shift becomes severe, and the weight MSE gets larger for all the methods. However, our proposed method keeps the weight MSE smaller than 0.8 for all the $\mu_t$ values, while KMM and L2IWE have the weight MSE more than 1.5. Also, for the prediction performance, our proposed method is close to the Oracle-Adaptation when $\mu_t$ increases with improved accuracy by more than 20%. But KMM and L2IWE generally get worse performance as $\mu_t$ increases.

We also evaluate the impact of target data size $m$ on the performance. We fix the source data size $n = 500$, and the target shift $\mu_t = 0.5$. We tune the size ratio $m/n$ in {0.2, 0.4, 0.6, 0.8}. The experimental results are shown in Figure 2 (c)&(f). It can be seen from Figure 2 (c) that the target data size $m$ has a relatively small impact on the adaptation performances. The weight MSE and delta accuracy curves are flat over different values of $m/n$. But when comparing the performance of our methods with KMM and L2IWE, we can see that our method can consistently achieve higher prediction accuracy and smaller weight MSE. Thus, we conclude that our proposed method performs better than the two existing methods under different sample sizes and target shift settings in this nonlinear regression study.

## 5.2 REAL-WORLD DATA EXPERIMENTS

In this section, we will conduct the experiments with two real-world datasets: the SOCR dataset[1] and the Szeged weather record dataset[2]. The SOCR dataset contains 1035 records of heights, weights, and position information for some current and recent Major League Baseball (MLB) players. In this dataset, there is a strong correlation between the player's weights and heights. Also, it is natural to consider the causal relation: height → weight, which justifies the continuous target shift assumption.

---

[1] http://wiki.stat.ucla.edu/socr/index.php/SOCR_Data_MLB_HeightsWeights
[2] https://www.kaggle.com/datasets/budincsevity/szeged-weather

Table 1: The experimental results for SOCR and Szeged weather datasets. The numbers reported before and after $\pm$ symbolize the mean and standard deviation, respectively.

| | SOCR Dataset | | Szeged Weather Dataset | |
|---|---|---|---|---|
| Adaption | Weight MSE | $\Delta$Pred. MSE (%) | Weight MSE | $\Delta$Pred. MSE (%) |
| Oracle | – | $12.726_{\pm 1.026}$ | - | $35.572_{\pm 26.237}$ |
| KMM | $0.477_{\pm 0.269}$ | $3.076_{\pm 5.175}$ | $2.343_{\pm 1.488}$ | $0.516_{\pm 65.107}$ |
| L2IWE | $0.178_{\pm 0.025}$ | $-5.933_{\pm 1.763}$ | $1.047_{\pm 0.650}$ | $-8.400_{\pm 27.622}$ |
| Ours | $0.025_{\pm 0.009}$ | $6.705_{\pm 0.977}$ | $0.756_{\pm 0.878}$ | $19.567_{\pm 15.679}$ |

Thus, we consider players' weights as the covariate $x$ and their heights as $y$. In our study, we conducted 20 random trials. In each trial, we treat all outfielder players as the testing data and randomly select 80% of the players with the other positions as the training source data.

The second real-world task under investigation is the Szeged temperature prediction problem. This dataset comprises weather-related data recorded in Szeged, Hungary, from 2006 to 2016. In this study, we take the temperature as the target, and humidity as covariate. This is because there is a causal relationship between temperature and humidity: it tends to decrease relative humidity when temperature rises. We treat each year's data as an individual trial. In each trial, we utilize data from January to October as the training source dataset, while data from November and December constitute the testing dataset. In this data-splitting scheme, the testing data tends to have lower temperature values than the training source data.

The experimental results are summarized in Table 1. As we do not know the true distribution of target variables in real-world datasets, we calculate the oracle importance weight in an empirical way: we use the real target values across training and testing datasets to estimate the corresponding density functions, then leverage the ratio of the estimated density functions for the oracle importance weight. We use weight MSE and $\Delta$prediction MSE (%) as the performance metrics. It can be seen that our proposed adaptation method performs better than KMM and L2IWE with smaller weight MSE and larger $\Delta$prediction MSE: Our method improves the weight estimation by reducing about 95% and 77% weight MSE for SOCR and Szeged weather datasets, respectively. Also, for $\Delta$prediction MSE, our method improves about 4% for the SOCR dataset and 19% for the Szeged weather dataset. Also, note that our method's estimation variation (i.e., the standard deviation) is smaller than KMM and IL2IWE. Thus, we claim that our method performs well in both estimation accuracy and stability.

## 6   DISCUSSIONS

We address the continuous target shift problem by nonparametrically estimating the importance weight function, by solving an ill-posed integral equation. The proposed method does not require distribution matching and only involves solving a linear system and has outperformed existing methods in experiments. Furthermore, we offer theoretical justification for the proposed methodology. Our approach has effectively extended to high-dimensional data by integrating the black-box-shift-estimation method introduced in Lipton et al. (2018). The essence of this method is to use a pre-trained predictive model to map the features $\mathbf{x}$ to a scalar $\widehat{E}_s(y|\mathbf{x})$; thus avoiding the curse of dimensionality. Due to space constraints, we present additional experiments on high-dimensional data in the supplementary materials.

This study is primarily centered on training predictive models in the target domain. An associated question naturally arises regarding the quantification of uncertainty in predictions. While existing research has addressed this concern in the context of classification (e.g., Podkopaev & Ramdas 2021), a noticeable gap exists when considering the continuous case.To our best knowledge, this aspect remains unexplored in the current literature and could be a prospective avenue for future research. Finally, it is noteworthy that we consider the case where the support of $y$ on the target domain is a subset of that in the source domain. Consequently, another potential future research question will be to address continuous target shifts under the out-of-distribution setting. Lastly, we adopt a simple density ratio estimation method (for $\eta(\mathbf{x})$) in this paper and establish theoretical results upon it. We compare this simple density ratio method with the RuLSIF method in Section C.2.3 of the supplementary, but more extensive comparisons with other existing methods could be of interest.

ACKNOWLEDGEMENT

Qinglong Tian is supported by the Natural Sciences and Engineering Research Council of Canada (NSERC) Discovery Grant RGPIN-2023-03479.

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
