# Supplementary Materials to "ReTaSA: A Nonparametric Functional Estimation Approach for Addressing Continuous Target Shift"

**Hwanwoo Kim**[1†]**, Xin Zhang**[2†]**, Jiwei Zhao**[3]**, Qinglong Tian**[4*]

University of Chicago[1], Meta, Inc.[2], University of Wisconsin-Madison[3], University of Waterloo[4]

## ABSTRACT

Section A provides detailed explanations on why the integral equation (1) has such "explosive" behavior and why the Tikhonov regularization can help alleviate this problem. Section B provides all the lemmas and proofs for establishing consistency and convergence rate. Section C contains extra numerical studies.

## A    ILLPOSEDNESS & TIKHONOV REGULARIZATION

Even if small perturbations in $\eta(\mathbf{x})$ can lead to large errors in the solution $\rho(y)$ in the integral equation (4). What exacerbates the problem in the target shift context is that we often do not know $\eta$ or $T$ but have to use estimates to replace them in (4), thus introducing a substantial amount of errors in the inputs. Omitting technical details, we explain the cause of the "explosive" behavior by utilizing the singular value decomposition of operators $T$ and $T^*$ (Kress, 1999). The solution to (4) can be represented in the following form:

$$\rho = \sum_{i \geq 0} \frac{1}{\lambda_i} \langle \eta, \psi_i \rangle \varphi_i, \tag{S.1}$$

where

1. $\{\varphi_i(y)\}$, $i \geq 0$ is an orthonormal sequence in $L^2(Y)$,
2. $\{\psi_i(\mathbf{x})\}$, $i \geq 0$ is an orthonormal sequence in $L^2(\mathbf{X})$,
3. $\{\lambda_i\}$, $i \geq 0$ is a sequence of positive numbers and $\lim_{i \to \infty} \lambda_i = 0$.

Suppose we perturb the input function $\eta(\mathbf{x})$ by $\eta^\delta(\mathbf{x}) = \eta(\mathbf{x}) + \delta\psi_j(\mathbf{x})$, the solution becomes $\rho^\delta(y) = \rho(y) + \delta\varphi_j(y)/\lambda_j$. The ratio between the change in the solution and in the input is $\|\rho^\delta - \rho\|/\|\eta^\delta - \eta\| = 1/\lambda_j$, which can be explosive as $\lambda_j$ can be arbitrarily close to zero. In other words, relatively small errors in estimating $\eta(\mathbf{x})$ can result in huge changes in the solution $\rho(y)$.

The Tikhonov regularized solution is given by

$$\rho^\alpha = \sum_{i \geq 0} \frac{\lambda_i}{\lambda_i^2 + \alpha} \langle \eta, \psi_i \rangle \varphi_i. \tag{S.2}$$

Here $\alpha > 0$ is a fixed regularization parameter that can mitigate the explosive term $1/\lambda_i$ in (S.1), thus endowing the regularized estimator $\rho^\alpha$ with stability.

---

*Correspondence to qinglong.tian@uwaterloo.ca    † Equal Contribution

## A.1 The $\beta$-regularity space

The $\beta$-regularity space amounts to a condition of the relationship between the smoothness on the function $\rho(y)$ (i.e., rate of decay of its Fourier coefficients) and the degree of ill-posedness on the operator $T$ (i.e., rate of decay of its singular values). For example, suppose the operator is severely ill-posed (i.e., the rate of decay of the singular values of $T$ is fast); the condition basically says that we will also need the function $\rho(y)$ to be very smooth (rapidly decay). Otherwise, $\rho(y)$ is no longer in the $\beta$-regularity space.

To better understand the ill-posedness of the operator $T$, we provide an example using the Gaussian distribution. The singular value decomposition (SVD) of general conditional expectation operators is non-trivial compared with its counterpart for matrices. To provide some intuitive explanations, we focus on the exponential family. More specifically, the Gaussian distribution in our response. Suppose we can factorize a joint distribution $p(x, y)$ into $p(x|y)p(y)$, where

$$p(x|y) = \frac{1}{\sqrt{2\pi\sigma^2}} \exp\left(-\frac{(x-y)^2}{2\sigma^2}\right), \quad p(y) = \frac{1}{\sqrt{2\pi\sigma_0^2}} \exp\left(-\frac{y^2}{2\sigma_0^2}\right)$$

In fact, this joint distribution $p(x, y)$ corresponds to the model $x = y + \epsilon$, where the random variable $y \sim \text{Norm}(0, \sigma_0^2)$ is independent of the random Gaussian noise term $\epsilon \sim \text{Norm}(0, \sigma^2)$. Through SVD of the conditional expectation operator $T$ (associated with $p(x|y)$), the singular values have a closed form and are given by

$$\lambda_i = \left(\frac{\sigma_0^2}{\sigma_0^2 + \sigma^2}\right)^{i/2}.$$

We can see that the decay rate of $\lambda_i$ depends on the ratio $\sigma^2/\sigma_0^2$. Recall the model behind the joint distribution is $x = y + \epsilon$, where $\sigma^2$ is the variance of $\epsilon$ and can be seen as noise and $\sigma_0^2$ is the variance of $Y$ and can be seen as information. Thus, we can see that if the noise dominates the information (i.e., $\sigma^2 \gg \sigma_0^2$), the decay is fast. On the other hand, if the information dominates the noise (i.e., $\sigma_0^2 \gg \sigma^2$), the decay is slow. If $\beta$ is given, and the noise-to-information ratio is high, the $\beta$-regularity space becomes smaller. In other words, it becomes more difficult to identify functions in the high-noise setting. The reason for the resulting smaller space is that we require the Fourier coefficients to decay at a fast rate to ensure the series is finite.

Now, back to the general model $p(x, y)$, qualitative speaking, the singular values $\lambda_i$ of the operator $T$ describe how well we know about $x$ based on the information of $y$. If $y$ does not provide much information about $x$, or in other words, $x$ contains a lot of noise, the singular values decay fast, and we say the conditional expectation operator is severely ill-posed, and vice versa.

The $\beta$-regularization spaces are to characterize both the singular values of the conditional expectation operator $T$ and the Fourier coefficients of the function $\rho(\cdot)$. The appropriate value for $\beta$ also depends on both $T$ and $\rho(\cdot)$. A larger $\beta$ implies a smaller function space for $\rho(\cdot)$ to live in, given the operator $T$. Lastly, under our identifiability assumption, we would like to point out that the $\beta$-regularity space can also be viewed as the range of the operator $(T^*T)^{\beta/2}$.

## B Proofs of Theorem 1

First recall definitions and assumptions we make, which are again provided in below.

**Assumption 1.** *The joint source density $p_s(\mathbf{x}, y)$ and marginal source densities $p_s(\mathbf{x})$ and $p_s(y)$ satisfy*

$$\int \int \left\{\frac{p_s(\mathbf{x}, y)}{p_s(\mathbf{x})p_s(y)}\right\}^2 p_s(\mathbf{x})p_s(y)d\mathbf{x}dy < \infty.$$

**Definition 1.** *For $\beta > 0$, the $\beta$-regularity space of the compact operator $T$ is defined as $\Phi_\beta = \left\{ \rho \in L^2(y) \text{ such that } \sum_{i=1}^\infty \langle \rho, \varphi_i \rangle^2 / \lambda_i^{2\beta} < \infty \right\}$, where $\{\lambda_i\}_{i=1}^\infty$ are the decreasing sequence of nonzero singular values of $T$ and $\{\varphi_i\}_{i=1}^\infty$ is a set of orthonormal sequence in $L^2(y)$. In other words, for some orthonormal sequence $\{\phi_i\}_{i=1}^\infty \subset L^2(\mathbf{x})$, $T\phi_i = \lambda_i \varphi_i$ holds for all $i \in \mathbb{N}$.*

**Assumption 2.** *There exists a $\beta > 0$ such that the solution $\rho_0 \in \Phi_\beta$.*

**Assumption 3.** *We assume $\mathbf{x} \in [0,1]^p$, $y \in [0,1]$; and the $k$-times continuously differentiable densities $p_s(\mathbf{x}, y)$, $p_s(\mathbf{x})$ and $p_t(\mathbf{x})$ are all bounded away from zero on their supports, i.e., $p_s(\mathbf{x}, y), p_s(\mathbf{x}), p_t(\mathbf{x}) \in [\epsilon, \infty)$, for some $\epsilon > 0$.*

**Definition 2.** *With a bandwidth parameter $h$, a function $K_h : [0,1]^p \times [0,1] \to \mathbb{R}_{\geq 0}$ satisfying 1. $K_h(x,y) = 0$ if $x \notin [y-1, y] \cap \mathcal{C}$, where $\mathcal{C}$ is a compact interval independent of $y$ 2. $\sup_{x \in \mathcal{C}, y \in [0,1]} |K_h(x,y)| < \infty$. 3. $h^{-(q+1)} \int_{y-1}^y x^q K_h(x,y) dx = \begin{cases} 1 & \text{for } q = 0, \\ 0 & \text{for } 1 \leq q \leq \ell - 1, \end{cases}$ is referred as a univariate generalized kernel function of order $\ell$.*

**Assumption 4.** *Letting $\gamma = \min\{k, \ell\}$, the kernel bandwidth $h$, regularization parameter $\alpha$ and the source sample size $n$ satisfy $\lim_{\substack{h \to 0 \\ n \to \infty}} \log(n)/(nh^{p+1}) = 0$. Furthermore, the target sample size $m$ satisfy $\lim_{\substack{h \to 0 \\ m \to \infty}} \log(m)/(mh^p) = 0$.*

In order to prove Theorem 1, we first state necessary lemmas with its proof.

**Lemma 1.** *With a kernel bandwidth parameter $h$ and $\gamma = \min\{k, \ell\}$, we assume $||T - \widehat{T}||^2 = O_p\left(1/(nh^{p+1}) + h^{2\gamma}\right)$, $||T^* - \widehat{T}^*||^2 = O_p\left(1/(nh^{p+1}) + h^{2\gamma}\right)$, where $||\cdot||$ is an operator norm.*

*Proof.* This results follow directly from the Lemma B.2. in the appendix of Darolles et al. (2011) and the fact that Hilbert-Schmidt norm bounds the operator norm. $\square$

**Lemma 2.** *Under the assumptions 3 and 4, our generalized kernel function-based approximation of $p_s(\mathbf{x})$, $p_t(\mathbf{x})$ denoted by $\widehat{p}_s(\mathbf{x})$, $\widehat{p}_t(\mathbf{x})$ satisfies*

- $\sup_{\mathbf{x} \in [0,1]^p} |\widehat{p}_s(\mathbf{x}) - p_s(\mathbf{x})| = o_p(1)$

- $\sup_{\mathbf{x} \in [0,1]^p} |\widehat{p}_t(\mathbf{x}) - p_t(\mathbf{x})| = o_p(1)$

*Proof.* This results follow directly from the Lemma B.1.(i) in the appendix of Darolles et al. (2011). $\square$

**Lemma 3.** *Under the assumptions 3 and 4, let $\widehat{p}_s(\mathbf{x})$ be the generalized kernel-based approximation of the source distribution $p_s(\mathbf{x})$. Then,*

$$\frac{1}{\inf_{\mathbf{x} \in [0,1]^p} \widehat{p}_s(\mathbf{x})} = O_p(1).$$

*Proof.* Observe that

$$|p_s(\mathbf{x})| \leq |p_s(\mathbf{x}) - \widehat{p}_s(\mathbf{x})| + |\widehat{p}_s(\mathbf{x})| \Rightarrow |p_s(\mathbf{x})| - o_p(1) \leq |\widehat{p}_s(\mathbf{x})|,$$

by Lemma 2. Hence

$$\inf_{\mathbf{x} \in [0,1]^p} |p_s(\mathbf{x})| - o_p(1) \leq \inf_{\mathbf{x} \in [0,1]^p} |\widehat{p}_s(\mathbf{x})|.$$

For a sufficiently large $n$ to construct $\widehat{p}_s(\mathbf{x})$, we have

$$\frac{1}{\inf_{\mathbf{x} \in [0,1]^p} |\widehat{p}_s(\mathbf{x})|} \leq \frac{1}{\inf_{\mathbf{x} \in [0,1]^p} |p_s(\mathbf{x})| - o_p(1)} = \frac{1}{\inf_{\mathbf{x} \in [0,1]^p} |p_s(\mathbf{x})|} + o_p(1) = O_p(1),$$

where the first equality holds by the continuous mapping theorem, and the second equality follows from the fact that $p_s$ is bounded away from zero. $\qquad\square$

**Lemma 4.** *Under assumptions 3 and 4 ,*

$$\sup_{\mathbf{x}\in[0,1]^p}\left[p_t(\mathbf{x})-\frac{1}{h^p}\mathbb{E}_s\left\{K_{\mathbf{X},h}(\mathbf{x}-\mathbf{x}_i,\mathbf{x})\frac{p_t(\mathbf{x}_i)}{p_s(\mathbf{x}_i)}\right\}\right]^2=O(h^{2\gamma}),$$

*where $\mathbb{E}_s$ denotes the expectation with respect to the source distribution and $\gamma=\min\{k,\ell\}$.*

*Proof.* Notice

$$\frac{1}{h^p}\mathbb{E}_s\left\{K_{\mathbf{X},h}(\mathbf{x}-\mathbf{x}_i,\mathbf{x})\frac{p_t(\mathbf{x}_i)}{p_s(\mathbf{x}_i)}\right\}=\frac{1}{h^p}\int_{[0,1]^p}K_{\mathbf{X},h}(\mathbf{x}-u,\mathbf{x})p_t(u)du$$

$$=\int_{\prod_{i=1}^p[\frac{x_i-1}{h},\frac{x_i}{h}]}K_{\mathbf{X},h}(hu,\mathbf{x})p_t(\mathbf{x}-hu)du$$

Using the Taylor expansion of $p_t(\mathbf{x}-hu)$ centered at $\mathbf{x}$ , the above is equivalent to

$$\int_{\prod_{i=1}^p[\frac{x_i-1}{h},\frac{x_i}{h}]}K_{\mathbf{X},h}(hu,\mathbf{x})\left\{p_t(\mathbf{x})+(-h)\sum_{i=1}^p\frac{\partial p_t(\mathbf{x})}{\partial x_i}u_i+\cdots+\frac{1}{\gamma!}\sum_{i_1,\cdots,i_\gamma}\frac{\partial^\gamma p_t(\mathbf{x}^*)}{\partial x_{i_1}\cdots x_{i_\gamma}}(-h)^\gamma u_{i_1}\cdots u_{i_\gamma}\right\}du.$$

From the definition, boundedness assumption of the generalized kernel function, and continuous differentiability $p_t(\mathbf{x})$, we get

$$\sup_{\mathbf{x}\in[0,1]^p}\frac{1}{h^p}\mathbb{E}_s\left\{K_{\mathbf{X},h}(\mathbf{x}-\mathbf{x}_i,\mathbf{x})\frac{p_t(\mathbf{x}_i)}{p_s(\mathbf{x}_i)}\right\}=p_t(\mathbf{x})+O(h^\gamma),$$

from which the statement follows. $\qquad\square$

**Lemma 5.** *Under assumptions 3 and 4,*

$$\left|w(v)-\int_{[\frac{-v}{h},\frac{1-v}{h}]}K_{Y,h}(hu,v+hu)w(v+hu)du\right|=O(h^\gamma),$$

*where $w(y)=\rho_0(y)+1=\frac{p_t(y)}{p_s(y)}$.*

*Proof.* Using the Taylor expansion of $w$ centered at $v$, you have

$$\int_{[\frac{-v}{h},\frac{1-v}{h}]}K_{Y,h}(hu,v+hu)w(v+hu)du$$

$$=\int_{[\frac{-v}{h},\frac{1-v}{h}]}K_{Y,h}(hu,v+hu)\left\{w(v)+h\frac{\partial w(v)}{\partial v}u+\cdots+\frac{1}{\gamma!}\frac{\partial^\gamma w(v^*)}{\partial v^\gamma}h^\gamma u^\gamma\right\}du$$

$$=w(v)+O(h^\gamma),$$

where the last equality holds due to the definition, boundedness of the generalized kernel function, and continuous differentiability of $w$. $\qquad\square$

**Theorem 1.** *For $\beta \geq 2$, under Assumptions 1 - 4, one can show that $||\widehat{\rho}_\alpha - \rho_0||^2$ is of order*

$$O_p\left(\alpha^2 + \left(\frac{1}{nh^{p+1}} + h^{2\gamma}\right)\left(\frac{1}{nh^{p+1}} + h^{2\gamma} + 1\right)\alpha^{(\beta-2)} + \frac{1}{\alpha^2}\left(\frac{1}{nh^{p+1}} + h^{2\gamma} + 1\right)\left(\frac{1}{n} + h^{2\gamma} + \sqrt{\frac{\log m}{mh^p}}\left(h^\gamma + \sqrt{\frac{\log m}{mh^p}}\right)\right)\right).$$

*In particular, if $\max\left(h^{2\gamma}, \sqrt{\frac{\log(m)}{mh^p}}h^\gamma, \frac{\log(m)}{mh^p}\right) = o_p(\alpha^2)$ and $\lim_{\substack{\alpha \to 0 \\ n \to \infty}} n\alpha^2 = \infty$, $\widehat{\rho}_\alpha$ converges in probability to $\rho_0$ as $\alpha, h \to 0$, $n \to \infty$, and $m \to \infty$.*

*Proof.* Notice $\widehat{\rho}_\alpha - \rho_0 = R_1 + R_2 + R_3$ where

$$R_1 = \left(\alpha\mathrm{I} + \widehat{T}^*\widehat{T}\right)^{-1}\widehat{T}^*\widehat{\eta} - \left(\alpha\mathrm{I} + \widehat{T}^*\widehat{T}\right)^{-1}\widehat{T}^*\widehat{T}\rho$$

$$R_2 = \left(\alpha\mathrm{I} + \widehat{T}^*\widehat{T}\right)^{-1}\widehat{T}^*\widehat{T}\rho - \left(\alpha\mathrm{I} + T^*T\right)^{-1}T^*T\rho$$

$$R_3 = \left(\alpha\mathrm{I} + T^*T\right)^{-1}T^*T\rho - \rho.$$

We establish upper bounds on the norm of $R_1$, $R_2$, and $R_3$, so as to bound the norm of $\widehat{\rho}_\alpha - \rho_0$. First of all, based on proposition 3.2 of the Darolles et al. (2011), we have $||R_3||^2 = O(\alpha^{\min(\beta,2)})$. Secondly, notice that

$$R_2 = \left(\alpha\mathrm{I} + \widehat{T}^*\widehat{T}\right)^{-1}\left(\alpha\mathrm{I} + \widehat{T}^*\widehat{T} - \alpha\mathrm{I}\right)\rho_0 - \left(\alpha\mathrm{I} + T^*T\right)^{-1}\left(\alpha\mathrm{I} + T^*T - \alpha\mathrm{I}\right)\rho_0$$

$$= \left(\mathrm{I} - \alpha\left(\alpha\mathrm{I} + \widehat{T}^*\widehat{T}\right)^{-1}\right)\rho_0 - \left(\mathrm{I} - \alpha\left(\alpha\mathrm{I} + T^*T\right)^{-1}\right)\rho_0$$

$$= \alpha\left(\alpha\mathrm{I} + T^*T\right)^{-1}\rho_0 - \alpha\left(\alpha\mathrm{I} + \widehat{T}^*\widehat{T}\right)^{-1}\rho_0.$$

We can rewrite

$$-R_2 = \alpha\left(\left(\alpha\mathrm{I} + \widehat{T}^*\widehat{T}\right)^{-1} - \left(\alpha\mathrm{I} + T^*T\right)^{-1}\right)\rho_0$$

$$= \alpha\left(\alpha\mathrm{I} + \widehat{T}^*\widehat{T}\right)^{-1}\left(T^*T - \widehat{T}^*\widehat{T}\right)\left(\alpha\mathrm{I} + T^*T\right)^{-1}\rho_0$$

$$= B_1 + B_2,$$

where

$$B_1 = \alpha\left(\alpha\mathrm{I} + \widehat{T}^*\widehat{T}\right)^{-1}T^*\left(T - \widehat{T}\right)\left(\alpha\mathrm{I} + T^*T\right)^{-1}\rho_0,$$

$$B_2 = \alpha\left(\alpha\mathrm{I} + \widehat{T}^*\widehat{T}\right)^{-1}\left(T^* - \widehat{T}^*\right)\widehat{T}\left(\alpha\mathrm{I} + T^*T\right)^{-1}\rho_0.$$

Now we provide a bound on the square of the norm of each term in $B_1$ and $B_2$. From Darolles et al. (2011), it is known that

$$\left|\left|\left(\alpha\mathrm{I} + \widehat{T}^*\widehat{T}\right)^{-1}\right|\right|^2 = O_p\left(\frac{1}{\alpha^2}\right).$$

From the assumption 1 and Lemma 1, we have

$$||T||^2 = O(1),$$

$$||\widehat{T}||^2 \leq 2||\widehat{T} - T||^2 + 2||T||^2 = O_p\left(\frac{1}{nh^{p+1}} + h^{2\gamma} + 1\right).$$

In addition, note that

$$
\begin{aligned}
\alpha \left(\alpha \mathrm{I} + T^*T\right)^{-1} \rho_0 &= \left(\mathrm{I} - \mathrm{I} + \alpha \left(\alpha \mathrm{I} + T^*T\right)^{-1}\right) \rho_0 \\
&= \left(\mathrm{I} - (\alpha \mathrm{I} + T^*T)^{-1}(\alpha \mathrm{I} + T^*T - \alpha \mathrm{I})\right) \rho_0 \\
&= \left(\mathrm{I} - (\alpha \mathrm{I} + T^*T)^{-1} T^*T\right) \rho_0 \\
&= \rho_0 - (\alpha \mathrm{I} + T^*T)^{-1} T^*T \rho_0 \\
&= \rho_0 - \rho_\alpha.
\end{aligned}
$$

Again by assumption 2 and the proposition of 3.11 of the Carrasco et al. (2007), we have

$$
\left\| \alpha \left(\alpha \mathrm{I} + T^*T\right)^{-1} \rho_0 \right\| = O(\alpha^{\min(\beta,2)}).
$$

Therefore, we have

$$
\begin{aligned}
\|B_1\|^2 &\leq \left\| \left(\alpha \mathrm{I} + \widehat{T}^*\widehat{T}\right)^{-1} \right\|^2 \|T^*\|^2 \|T - \widehat{T}\|^2 \|\alpha \left(\alpha \mathrm{I} + T^*T\right)^{-1} \rho_0\|^2 \\
&= O_p\left( \left(\frac{1}{nh^{p+1}} + h^{2\gamma}\right) \alpha^{\min(\beta-2,0)} \right),
\end{aligned}
$$

$$
\begin{aligned}
\|B_2\|^2 &\leq \left\| \left(\alpha \mathrm{I} + \widehat{T}^*\widehat{T}\right)^{-1} \right\|^2 \|T^* - \widehat{T}^*\|^2 \|\widehat{T}\|^2 \|\alpha \left(\alpha \mathrm{I} + T^*T\right)^{-1} \rho_0\|^2 \\
&= O_p\left( \left(\frac{1}{nh^{p+1}} + h^{2\gamma}\right)^2 \alpha^{\min(\beta-2,0)} + \left(\frac{1}{nh^{p+1}} + h^{2\gamma}\right) \alpha^{\min(\beta-2,0)} \right).
\end{aligned}
$$

Combining all together, we get

$$
\|R_2\|^2 \leq 2\|B_1\|^2 + 2\|B_2\|^2 = O_p\left( \left(\frac{1}{nh^{p+1}} + h^{2\gamma}\right)^2 \alpha^{\min(\beta-2,0)} + \left(\frac{1}{nh^{p+1}} + h^{2\gamma}\right) \alpha^{\min(\beta-2,0)} \right).
$$

Lastly, we obtain the upper bound on the norm of $R_1$. Note that

$$
\|R_1\|^2 \leq \left\| \left(\alpha \mathrm{I} + \widehat{T}^*\widehat{T}\right)^{-1} \right\|^2 \|\widehat{T}^*\|^2 \|\widehat{\eta} - \widehat{T}\rho_0\|^2.
$$

Recall that $\left\| \left(\alpha \mathrm{I} + \widehat{T}^*\widehat{T}\right)^{-1} \right\|^2 = O_p\left(\frac{1}{\alpha^2}\right)$ and $\|T^*\|^2 = O_p\left(\frac{1}{nh^{p+1}} + h^{2\gamma} + 1\right)$ by assumption 1 and Lemma 1. To obtain an upper bound on $\|\widehat{\eta} - \widehat{T}\rho_0\|^2$, note that

$$
\begin{aligned}
\widehat{\eta} - \widehat{T}\rho_0 &= \widehat{\eta} + 1 - (\widehat{T}\rho_0 + 1) \\
&= \frac{\widehat{p}_t(\mathbf{x})}{\widehat{p}_s(\mathbf{x})} - \left( \int_{[0,1]} (\rho_0(y) + 1) \frac{\widehat{p}_s(\mathbf{x}, y)}{\widehat{p}_s(\mathbf{x})} dy \right) \\
&= \frac{1}{\widehat{p}_s(\mathbf{x})} \left( \widehat{p}_t(\mathbf{x}) - \int_{[0,1]} w(y) \widehat{p}_s(\mathbf{x}, y) dy \right),
\end{aligned}
$$

where $w(y) = \rho_0(y) + 1$. Here, $\widehat{p}_t(\mathbf{x})$ is the KDE estimator for $p_t(\mathbf{x})$ based on the unlabeled target data, and $\widehat{p}_s(\mathbf{x}), \widehat{p}_s(\mathbf{x}, y)$ are respectively the KDE estimator for $p_s(\mathbf{x}), p_s(\mathbf{x}, y)$ based on the source data. For succinct expressions, we introduce the following notation;

$$
A(\mathbf{x}) = \widehat{p}_t(\mathbf{x}) - \int_{[0,1]} w(y) \widehat{p}_s(\mathbf{x}, y) dy.
$$

One can observe that

$$\left\|\left\|\frac{A(\mathbf{x})}{\widehat{p}_s(\mathbf{x})} - \frac{A(\mathbf{x})}{p_s(\mathbf{x})}\right\|\right\| = \left\|\left\|\frac{A(\mathbf{x})p_s(\mathbf{x}) - A(\mathbf{x})\widehat{p}_s(\mathbf{x})}{\widehat{p}_s(\mathbf{x})p_s(\mathbf{x})}\right\|\right\| \leq \frac{\sup_{\mathbf{x} \in [0,1]^p} |p_s(\mathbf{x}) - \widehat{p}_s(\mathbf{x})|}{\inf_{\mathbf{x} \in [0,1]^p} |\widehat{p}_s(\mathbf{x})|} \left\|\left\|\frac{A(\mathbf{x})}{p_s(\mathbf{x})}\right\|\right\|.$$

By Lemma 2 and 3, we observe that

$$\left\|\left\|\frac{A(\mathbf{x})}{\widehat{p}_s(\mathbf{x})} - \frac{A(\mathbf{x})}{p_s(\mathbf{x})}\right\|\right\| = O_p\left(\left\|\left\|\frac{A(\mathbf{x})}{p_s(\mathbf{x})}\right\|\right\|\right),$$

which gives us

$$\left\|\left\|\frac{A(\mathbf{x})}{\widehat{p}_s(\mathbf{x})}\right\|\right\| = \left\|\left\|\frac{A(\mathbf{x})}{p_s(\mathbf{x})}\right\|\right\| + O_p\left(\left\|\left\|\frac{A(\mathbf{x})}{p_s(\mathbf{x})}\right\|\right\|\right) = O_p\left(\left\|\left\|\frac{A(\mathbf{x})}{p_s(\mathbf{x})}\right\|\right\|\right) \Rightarrow ||\widehat{\eta} - \widehat{T}\rho_0||^2 = O_p\left(\left\|\left\|\frac{A(\mathbf{x})}{p_s(\mathbf{x})}\right\|\right\|^2\right)$$

To establish an upperbound of $\left\|\left\|\frac{A(\mathbf{x})}{p_s(\mathbf{x})}\right\|\right\|^2$, we decompose $A(\mathbf{x})$ into

$$A(\mathbf{x}) = A_1(\mathbf{x}) + A_2(\mathbf{x}) + A_3(\mathbf{x}),$$

with

$$A_1(\mathbf{x}) = \widehat{p}_t(\mathbf{x}) - \frac{1}{nh^p} \sum_{i=1}^{n} K_{\mathbf{X},h}(\mathbf{x} - \mathbf{x}_i, \mathbf{x})\zeta(\mathbf{x}_i),$$

$$A_2(\mathbf{x}) = \frac{1}{nh^p} \sum_{i=1}^{n} K_{\mathbf{X},h}(\mathbf{x} - \mathbf{x}_i, \mathbf{x})\zeta(\mathbf{x}_i) - \frac{1}{nh^p} \sum_{i=1}^{n} K_{\mathbf{X},h}(\mathbf{x} - \mathbf{x}_i, \mathbf{x})w(y_i),$$

$$A_3(\mathbf{x}) = \frac{1}{nh^p} \sum_{i=1}^{n} K_{\mathbf{X},h}(\mathbf{x} - \mathbf{x}_i, \mathbf{x})w(y_i) - \int_{[0,1]} w(y)\widehat{p}_s(\mathbf{x}, y)dy$$

$$= \frac{1}{nh^p} \sum_{i=1}^{n} K_{\mathbf{X},h}(\mathbf{x} - \mathbf{x}_i, \mathbf{x})w(y_i) - \frac{1}{nh^{p+1}} \sum_{i=1}^{n} \int_{[0,1]} w(y)K_{Y,h}(y - y_i, y)K_{\mathbf{X},h}(\mathbf{x} - \mathbf{x}_i, \mathbf{x})dy$$

$$= \frac{1}{nh^p} \sum_{i=1}^{n} K_{\mathbf{X},h}(\mathbf{x} - \mathbf{x}_i, \mathbf{x})\left(w(y_i) - \frac{1}{h}\int_{[0,1]} w(y)K_{Y,h}(y - y_i, y)dy\right),$$

where $(\mathbf{x}_i, y_i)_{i=1}^{n}$ are i.i.d. samples of $(\mathbf{X}, Y) \sim p_s$, $\zeta(\mathbf{x}) = \frac{p_t(\mathbf{x})}{p_s(\mathbf{x})}$ and $w(y) = \rho_0(y) + 1 = \frac{p_t(y)}{p_s(y)}$. Here, $K_{\mathbf{X},h}$ is the product of $p$ univariate generalized kernel function of of order $\ell$. In a similar fashion, $K_{Y,h}$ is a univariate generalized kernel function of of order $\ell$. . First, we establish the upper bound of $\left\|\left\|\frac{A_1(\mathbf{x})}{p_s(\mathbf{x})}\right\|\right\|^2$. We re-express $A_1(\mathbf{x})$ into

$$A_1(\mathbf{x}) = \widehat{p}_t(\mathbf{x}) - p_t(\mathbf{x}) + p_t(\mathbf{x}) - \frac{1}{nh^p} \sum_{i=1}^{n} K_{\mathbf{X},h}(\mathbf{x} - \mathbf{x}_i, \mathbf{x})\zeta(\mathbf{x}_i) = A_{11}(\mathbf{x}) + A_{12}(\mathbf{x}),$$

where

$$A_{11}(\mathbf{x}) = \widehat{p}_t(\mathbf{x}) - p_t(\mathbf{x})$$

$$A_{12}(\mathbf{x}) = p_t(\mathbf{x}) - \frac{1}{nh^p} \sum_{i=1}^{n} K_{\mathbf{X},h}(\mathbf{x} - \mathbf{x}_i, \mathbf{x})\zeta(\mathbf{x}_i).$$

By the Lemma B.1. in the appendix of Darolles et al. (2011), we observe that

$$\left\|\frac{A_{11}(\mathbf{x})}{p_s(\mathbf{x})}\right\|^2 \leq \sup_{\mathbf{x}\in[0,1]^p} |\widehat{p}_t(\mathbf{x}) - p_t(\mathbf{x})|^2 \left(\int_{[0,1]^p} \frac{1}{p_s(\mathbf{x})} d\mathbf{x}\right) = O_p\left(\sqrt{\frac{\log m}{mh^p}}\left(h^\gamma + \sqrt{\frac{\log m}{mh^p}}\right) + h^{2\gamma}\right)$$

Next, we have

$$A_{12}(\mathbf{x})^2 = p_t(\mathbf{x})^2 - \frac{2p_t(\mathbf{x})}{nh^p}\sum_{i=1}^n K_{\mathbf{X},h}(\mathbf{x} - \mathbf{x}_i, \mathbf{x})\zeta(\mathbf{x}_i) +$$

$$\frac{1}{n^2 h^{2p}}\left(\sum_{i,j=1}^n K_{\mathbf{X},h}(\mathbf{x} - \mathbf{x}_i, \mathbf{x})K_{\mathbf{X},h}(\mathbf{x} - \mathbf{x}_j, \mathbf{x})\zeta(\mathbf{x}_i)\zeta(\mathbf{x}_j)\right),$$

which leads to

$$\mathbb{E}_s\left(A_{12}(\mathbf{x})^2\right) = p_t(\mathbf{x})^2 - \frac{2p_t(\mathbf{x})}{h^p}\mathbb{E}_s\left(K_{\mathbf{X},h}(\mathbf{x} - \mathbf{x}_i, \mathbf{x})\zeta(\mathbf{x}_i)\right) + \frac{n-1}{nh^{2p}}\mathbb{E}_s^2\left(K_{\mathbf{X},h}(\mathbf{x} - \mathbf{x}_i, \mathbf{x})\zeta(\mathbf{x}_i)\right)$$

$$+ \frac{1}{nh^{2p}}\mathbb{E}_s\left(K_{\mathbf{X},h}(\mathbf{x} - \mathbf{x}_i, \mathbf{x})^2\zeta(\mathbf{x}_i)^2\right)$$

$$\leq \left(p_t(\mathbf{x}) - \frac{1}{h^p}\mathbb{E}_s\left(K_{\mathbf{X},h}(\mathbf{x} - \mathbf{x}_i, \mathbf{x})\zeta(\mathbf{x}_i)\right)\right)^2 + \frac{1}{nh^{2p}}\mathbb{E}_s\left(K_{\mathbf{X},h}(\mathbf{x} - \mathbf{x}_i, \mathbf{x})^2\zeta(\mathbf{x}_i)^2\right),$$

where the expectation is with respect to the source distribution. By the Fubini's theorem, we have

$$\mathbb{E}_s\left(\left\|\frac{A_{12}(\mathbf{x})}{p_s(\mathbf{x})}\right\|^2\right) = \int_{[0,1]^p} \mathbb{E}_s\left(A_{12}(\mathbf{x})^2\right)\frac{1}{p_s(\mathbf{x})}d\mathbf{x}.$$

Note that, using the change of variables,

$$\frac{1}{nh^{2p}}\mathbb{E}_s\left(K_{\mathbf{X},h}(\mathbf{x} - \mathbf{x}_i, \mathbf{x})^2\zeta(\mathbf{x}_i)^2\right) = \frac{1}{n}\int_{[0,1]^p}\frac{1}{h^{2p}}K_{\mathbf{X},h}(\mathbf{x} - u, \mathbf{x})^2\frac{p_t(u)^2}{p_s(u)}du$$

$$= \frac{1}{n}\int_{\prod_{i=1}^p[\frac{x_i-1}{h^2}, \frac{x_i}{h^2}]} K_{\mathbf{X},h}(h^2 u, \mathbf{x})^2\frac{p_t(\mathbf{x} - h^2 u)^2}{p_s(\mathbf{x} - h^2 u)}du.$$

Boundedness of $K_{\mathbf{X},h}$, continuity and nonzero lower bound of $p_t$ and $p_s$ yields

$$\frac{1}{nh^{2p}}\mathbb{E}_s\left(K_{\mathbf{X},h}(\mathbf{x} - \mathbf{x}_i, \mathbf{x})^2\zeta(\mathbf{x}_i)^2\right) = O\left(\frac{1}{n}\right).$$

Combining with the Lemma 4, which states that

$$\sup_{\mathbf{x}\in[0,1]^p}\left(p_t(\mathbf{x}) - \frac{1}{h^p}\mathbb{E}_s\left(K_{\mathbf{X},h}(\mathbf{x} - \mathbf{x}_i, \mathbf{x})\zeta(\mathbf{x}_i)\right)\right)^2 = O(h^{2\gamma}),$$

we deduce that

$$\mathbb{E}_s\left(\left\|\frac{A_{12}(\mathbf{x})}{p_s(\mathbf{x})}\right\|^2\right) = O\left(h^{2\gamma} + \frac{1}{n}\right),$$

which implies

$$\left\|\frac{A_{12}(\mathbf{x})}{p_s(\mathbf{x})}\right\|^2 = O_p\left(h^{2\gamma} + \frac{1}{n}\right),$$

by the Markov inequality. Combining the upperbound of $\left|\left|\frac{A_{11}(\mathbf{x})}{p_s(\mathbf{x})}\right|\right|^2$ and $\left|\left|\frac{A_{12}(\mathbf{x})}{p_s(\mathbf{x})}\right|\right|^2$, we have

$$\left|\left|\frac{A_1(\mathbf{x})}{p_s(\mathbf{x})}\right|\right|^2 = O_p\left(\sqrt{\frac{\log m}{mh^p}}\left(h^\gamma + \sqrt{\frac{\log m}{mh^p}}\right) + h^{2\gamma} + \frac{1}{n}\right) \tag{S.3}$$

Next, we establish an upperbound for $\left|\left|\frac{A_2(\mathbf{x})}{p_s(\mathbf{x})}\right|\right|^2$. Let $\Delta_i = \zeta(\mathbf{x}_i) - w(y_i)$. Then,

$$\begin{aligned}
\mathbb{E}_s\left(\left|\left|\frac{A_2(\mathbf{x})}{p_s(\mathbf{x})}\right|\right|^2\right) &= \mathbb{E}_s\left(\left|\left|\frac{1}{nh^p p_s(\mathbf{x})}\sum_{i=1}^n K_{\mathbf{X},h}(\mathbf{x} - \mathbf{x}_i, \mathbf{x})\Delta_i\right|\right|^2\right) \\
&= \frac{1}{n^2 h^{2p}}\sum_{i\neq j}\int_{[0,1]^p}\mathbb{E}_s(\Delta_i K_{\mathbf{X},h}(\mathbf{x} - \mathbf{x}_i, \mathbf{x}))\mathbb{E}_s(\Delta_j K_{\mathbf{X},h}(\mathbf{x} - \mathbf{x}_j, \mathbf{x}))\frac{1}{p_s(\mathbf{x})}d\mathbf{x} \\
&\quad + \frac{1}{nh^{2p}}\int_{[0,1]^p}\mathbb{E}_s(\Delta_i^2 K_{\mathbf{X},h}^2(\mathbf{x} - \mathbf{x}_i, \mathbf{x}))\frac{1}{p_s(\mathbf{x})}d\mathbf{x}
\end{aligned}$$

where in the the last equality we used the Fubini's theorem. Now one can show that

$$\mathbb{E}_s(\Delta_i K_{\mathbf{X},h}(\mathbf{x} - \mathbf{x}_i, \mathbf{x})) = \mathbb{E}_{\mathbf{x}_i}\left(\mathbb{E}_{y_i|\mathbf{x}_i}(\Delta_i K_{\mathbf{X},h}(\mathbf{x} - \mathbf{x}_i, \mathbf{x}))\right) = \mathbb{E}_{\mathbf{x}_i}\left(\mathbb{E}_{y_i|\mathbf{x}_i}(\Delta_i)K_{\mathbf{X},h}(\mathbf{x} - \mathbf{x}_i, \mathbf{x})\right) = 0$$

since

$$\begin{aligned}
\mathbb{E}_{y_i|\mathbf{x}_i}(\Delta_i) &= \zeta(\mathbf{x}_i) - \int_{[0,1]}\frac{p_t(y)}{p_s(y)}p_s(y|\mathbf{x}_i)dy = \zeta(\mathbf{x}_i) - \int_{[0,1]}\frac{p_t(\mathbf{x}_i, y)}{p_s(\mathbf{x}_i, y)}p_s(y|\mathbf{x}_i)dy \\
&= \zeta(\mathbf{x}_i) - \frac{p_t(\mathbf{x}_i)}{p_s(\mathbf{x}_i)}\int_{[0,1]}p_t(y|\mathbf{x}_i)dy = 0.
\end{aligned}$$

Notice that we used the fact $\frac{p_t(y)}{p_s(y)} = \frac{p_t(\mathbf{x}_i, y)}{p_s(\mathbf{x}_i, y)}$ in the first equality above, which is true by the target shift assumption. Furthermore,

$$\begin{aligned}
&\frac{1}{h^{2p}}\mathbb{E}_s(\Delta_i^2 K_{\mathbf{X},h}^2(\mathbf{x} - \mathbf{x}_i, \mathbf{x})) \\
&= \int_{[0,1]}\int_{[0,1]^p}\frac{1}{h^{2p}}\left(\zeta(u) - w(y)\right)^2 K_{\mathbf{X},h}^2(\mathbf{x} - u, \mathbf{x})p_s(u, y)dudy \\
&= \int_{[0,1]}\int_{\prod_{i=1}^p[\frac{x_i-1}{h^2}, \frac{x_i}{h^2}]}\left(\zeta(\mathbf{x} - h^2 u) - w(y)\right)^2 K_{\mathbf{X},h}^2(h^2 u, \mathbf{x})p_s(\mathbf{x} - h^2 u, y)dudy.
\end{aligned}$$

By the boundedness of $K_{\mathbf{X},h}, p_s, p_t$ and positive lowerbound of $p_s, p_t$, we observe that

$$\frac{1}{h^{2p}}\int_{[0,1]^p}\mathbb{E}_s(\Delta_i^2 K_{\mathbf{X},h}^2(\mathbf{x} - \mathbf{x}_i, \mathbf{x}))\frac{1}{p_s(\mathbf{x})}d\mathbf{x} = O(1).$$

Combining everything, we get $\mathbb{E}_s\left(\left|\left|\frac{A_2(\mathbf{x})}{p_s(\mathbf{x})}\right|\right|^2\right) = O\left(\frac{1}{n}\right)$. Again using the Markov's inequality this leads to

$$\left|\left|\frac{A_2(\mathbf{x})}{p_s(\mathbf{x})}\right|\right|^2 = O_p\left(\frac{1}{n}\right).$$

Lastly, to obtain the upperbound of $\left|\left|\frac{A_3(\mathbf{x})}{p_s(\mathbf{x})}\right|\right|^2$, notice

$$\mathbb{E}_s\left|\left|\frac{A_3(\mathbf{x})}{p_s(\mathbf{x})}\right|\right|^2 = \int_{[0,1]^p}\mathbb{E}_s\left(A_3^2(\mathbf{x})\right)\frac{1}{p_s(\mathbf{x})}d\mathbf{x} = O\left(\sup_{\mathbf{x}\in[0,1]^p}\mathbb{E}_s\left(A_3^2(\mathbf{x})\right)\right).$$

Letting

$$B(y_i) = w(y_i) - \frac{1}{h}\int_{[0,1]} w(y)K_{Y,h}(y - y_i, y)dy,$$

we have

$$A_3(\mathbf{x}) = \frac{1}{nh^p}\sum_{i=1}^{n} K_{\mathbf{X},h}(\mathbf{x} - \mathbf{x}_i, \mathbf{x})B(y_i).$$

Therefore,

$$\mathbb{E}_s\left(A_3^2(\mathbf{x})\right) = \mathbb{E}_s\left(\frac{1}{n^2h^{2p}}\Big(\sum_{i=1}^{n} K_{\mathbf{x},h}^2(\mathbf{x} - \mathbf{x}_i, \mathbf{x})B^2(y_i)\right.$$

$$\left.+ \sum_{i\neq j} K_{\mathbf{X},h}(\mathbf{x} - \mathbf{x}_i, \mathbf{x})K_{\mathbf{X},h}(\mathbf{x} - \mathbf{x}_j, \mathbf{x})B(y_i)B(y_j)\Big)\right)$$

$$= \frac{1}{nh^{2p}}\mathbb{E}_s(K_{\mathbf{X},h}^2(\mathbf{x} - \mathbf{x}_i, \mathbf{x})B^2(y_i)) + \frac{n-1}{nh^{2p}}\mathbb{E}_s^2(K_{\mathbf{X},h}(\mathbf{x} - \mathbf{x}_i, \mathbf{x})B(y_i))$$

$$\leq \frac{1}{nh^{2p}}\mathbb{E}_s(K_{\mathbf{X},h}^2(\mathbf{x} - \mathbf{x}_i, \mathbf{x})B^2(y_i)) + \frac{1}{h^{2p}}\mathbb{E}_s^2(K_{\mathbf{X},h}(\mathbf{x} - \mathbf{x}_i, \mathbf{x})B(y_i)).$$

Using change of variables,

$$\frac{1}{h^{2p}}\mathbb{E}_s(K_{\mathbf{X},h}^2(\mathbf{x} - \mathbf{x}_i, \mathbf{x})B^2(y_i))$$

$$= \int_{[0,1]}\int_{[0,1]^p} \frac{1}{h^{2p}}K_{\mathbf{X},h}^2(\mathbf{x} - u, \mathbf{x})\left(w(v) - \frac{1}{h}\int_{[0,1]} K_{Y,h}(y - v, y)w(y)dy\right)^2 p_s(u, v)dudv$$

$$= \int_{[0,1]}\int_{\prod_{i=1}^{p}[\frac{x_i-1}{h^2}, \frac{x_i}{h^2}]} K_{\mathbf{X},h}^2(h^2 u, \mathbf{x})\left(w(v) - \frac{1}{h}\int_{[0,1]} K_{Y,h}(y - v, y)w(y)dy\right)^2 p_s(\mathbf{x} - h^2 u, v)dudv$$

$$= O\left(\int_{[0,1]}\left(w(v) - \frac{1}{h}\int_{[0,1]} K_{Y,h}(y - v, y)w(y)dy\right)^2 dv\right),$$

where the last equality holds for any $\mathbf{x}$ due to boundedness of $K_{\mathbf{x},h}$ and $p_s$ with a positive lower bound of $p_s$. In particular, we have

$$\sup_{\mathbf{x}\in[0,1]^p} \frac{1}{h^{2p}}\mathbb{E}_s(K_{\mathbf{X},h}^2(\mathbf{x} - \mathbf{x}_i, \mathbf{x})B^2(y_i)) = O\left(\int_{[0,1]}\left(w(v) - \frac{1}{h}\int_{[0,1]} K_{Y,h}(y - v, y)w(y)dy\right)^2 dv\right).$$

We observe that

$$w(v) - \frac{1}{h}\int_{[0,1]} K_{Y,h}(y - v, y)w(y)dy = w(v) - \int_{[\frac{-v}{h}, \frac{1-v}{h}]} K_{Y,h}(hu, v + hu)w(v + hu)du,$$

and,

$$\int_{[0,1]}\left(w(v) - \int_{[\frac{-v}{h}, \frac{1-v}{h}]} K_{Y,h}(hu, v + hu)w(v + hu)du\right)^2 dv$$

$$\leq 2\left(\int_{[0,1]} w^2(v)dv + \int_{[0,1]}\left(\int_{[\frac{-v}{h}, \frac{1-v}{h}]} K_{Y,h}(hu, v + hu)w(v + hu)du\right)^2 dv\right)$$

by the boundedness of $K_{Y,h}$, $p_s$, $p_t$ and the positive lower-bounds of $p_s$, $p_t$, we know that the quantity above is bounded and hence,

$$\sup_{\mathbf{x} \in [0,1]^p} \frac{1}{nh^{2p}} \mathbb{E}_s(K_{\mathbf{x},h}^2(\mathbf{x} - \mathbf{x}_i, \mathbf{x}) B^2(y_i)) = O\left(\frac{1}{n}\right).$$

To deal with the remaining term, one can observe that

$$\sup_{\mathbf{x} \in [0,1]^p} \left| \frac{1}{h^p} \mathbb{E}_s(K_{\mathbf{X},h}(\mathbf{x} - \mathbf{x}_i, \mathbf{x}) B(y_i)) \right|$$

$$\leq \sup_{\mathbf{x} \in [0,1]^p} \int_{[0,1]} \int_{[0,1]^p} \left| \frac{1}{h^p} K_{\mathbf{X},h}(\mathbf{x} - u, \mathbf{x}) \left( w(v) - \frac{1}{h} \int_{[0,1]} K_{Y,h}(y - v, y) w(y) dy \right) p_s(u,v) \right| du dv$$

$$= \sup_{\mathbf{x} \in [0,1]^p} \int_{[0,1]} \int_{[0,1]^p} \left| K_{\mathbf{X},h}(hu, \mathbf{x}) \left( w(v) - \frac{1}{h} \int_{[0,1]} K_{Y,h}(y - v, y) w(y) dy \right) p_s(\mathbf{x} - hu, v) \right| du dv$$

$$= O\left( \left| w(v) - \frac{1}{h} \int_{[0,1]} K_{Y,h}(y - v, y) w(y) dy \right| \right)$$

$$= O\left( \left| w(v) - \int_{[\frac{-v}{h}, \frac{1-v}{h}]} K_{Y,h}(hu, v + hu) w(v + hu) du \right| \right)$$

where the second to the last equality comes from the boundedness of $K_{\mathbf{X},h}$, $p_s$ and the positive lower-bounds of $p_s$. From the Lemma 5, we know that

$$\sup_{\mathbf{x} \in [0,1]^p} \frac{1}{h^p} \mathbb{E}_s(|K_{\mathbf{x},h}(\mathbf{x} - \mathbf{x}_i, \mathbf{x}) B(y_i)|) = O(h^\gamma).$$

Therefore, we get

$$\sup_{\mathbf{x} \in [0,1]^p} \mathbb{E}_s\left(A_3^2(\mathbf{x})\right) = O\left(\frac{1}{n} + h^{2\gamma}\right) \Rightarrow \sup_{\mathbf{x} \in [0,1]^p} \mathbb{E}_s \left\| \frac{A_3(\mathbf{x})}{p_s(\mathbf{x})} \right\|^2 = O\left(\frac{1}{n} + h^{2\gamma}\right)$$

$$\Rightarrow \left\| \frac{A_3(\mathbf{x})}{p_s(\mathbf{x})} \right\|^2 = O_p\left(\frac{1}{n} + h^{2\gamma}\right)$$

Combining together, we establish the Theorem 1. □

### B.1 DISCUSSION ON TARGET DOMAIN GENERALIZATION ERROR

The target-domain generalization error is given by

$$\mathbb{E}_s\left\{ \omega(y) \ell(\widehat{f}(\mathbf{x}), y) \right\} - \frac{1}{n} \sum_{i=1}^n \widehat{\omega}(y_i) \ell(\widehat{f}(\mathbf{x}_i), y_i).$$

One approach to bound this generalization error is by uniformly bounding the difference between the true and empirical risks over the hypothesis class of models.

$$\sup_f \left| \frac{1}{n} \sum_{i=1}^n \widehat{\omega}(y_i) \ell(f(\mathbf{x}_i, y_i)) - \mathbb{E}_s\left\{ \omega(y) \ell(f(\mathbf{x}), y) \right\} \right|.$$

Under some regularity conditions, the error above is dominated by

$$\sup_f |\mathbb{E}_s \{\omega(y)\ell(f(\mathbf{x}), y)\} - \mathbb{E}_s \{\widehat{\omega}(y)\ell(f(\mathbf{x}), y)\}| \leq \sqrt{\mathbb{E}_s \{\omega(y) - \widehat{\omega}(y)\}^2 \sup_f \mathbb{E}_s \{\ell^2(f(\mathbf{x}), y)\}}.$$

For the term that involves $\widehat{\omega}$, we can see that we can bound the target domain error by directly bounding $\mathbb{E}_s \{\omega(y) - \widehat{\omega}(y)\}^2$. In Theorem 1, we provide a bound for

$$\|\widehat{\rho} - \rho_0\|^2 = \int \{\widehat{\rho}(y) - \rho_0(y)\}^2 p_s(y) dy = \mathbb{E}_s \{\omega(y) - \widehat{\omega}(y)\}^2.$$

Thus, the weighted $L^2$ norm we are using is directly bounding the target domain generalization error.

## C   EXPERIMENTAL DETAILS AND RESULTS

### C.1   DETAILS FOR EXPERIMENTS IN SECTION 5

In this section, we will provide the details for experimental studies in Section 5.

**Dataset and Model Setup.**

1. Synthetic nonlinear regression dataset: the data are generated with $x = y + 3 \times \tanh(y) + \epsilon$, where $\epsilon \sim \text{Normal}(0, 1)$. The training source data have target $y_s$ from $\text{Normal}(0, 2)$ and the testing data have the target $y_t$ from $\text{Normal}(\mu_t, 0.5)$. $\mu_t$ is used to adjust the target shift. As for the target prediction, we adopt the polynomial regression model with degrees as 5: $y = \beta_0 + \sum_{i=1}^{5} \beta_i x^i$.

2. The SOCR dataset[1] contains 1035 records of heights, weights, and position information for some current and recent Major League Baseball (MLB) players. This task aims to predict the players' heights with their weights. For the data splitting, we use the data of outfielder players as the testing data and the others as the training source data. Thus, there are a total of 841 records in the training source data and 194 records testing data. As shown in Figure S.1(a), there is a strong correlation between the player's weights and heights. Also, it is natural to consider the causal relation: height $\rightarrow$ weight, which justifies the continuous target shift assumption. Figure S.1(b)-(c) show the target shift between the training and testing data. It can be seen that the testing data (i.e., the outfielder players) tend to have lower height values compared with the training source data. For the target prediction, we adopt the spline regression model.

3. The Szeged weather dataset[2] comprises weather-related data recorded in Szeged, Hungary, from 2006 to 2016. Our study uses the noon temperature as the target and treats the humidity as the covariate. We utilize data from January to October as the training source dataset, while data from November and December constitute the testing dataset, so the testing data tend to have lower temperature values than the training source data. This is because there is a causal relationship between temperature and humidity: it tends to decrease relative humidity when temperature rises (also see Figure S.2(a).

**Hyperparameter Tuning.** Our proposed adaptation method involves two categories of hyperparameters: kernel bandwidths and a regularization parameter. We use the kernel bandwidths for the first category to construct the matrices $\mathbf{C}_\xi$ and $\mathbf{C}_\zeta$. We adopt the median trick, as proposed in the works of Zhang et al. (2013); Garreau et al. (2017), to select the bandwidths. Specifically, for $\mathbf{C}_\xi$, we set the bandwidth $\nu_\xi = \frac{1}{2}\sqrt{H_{\xi,\tilde{n}}/2}$, where $H_{\xi,\tilde{n}} = \text{median}(\|\xi_i - \xi_j\|^2, 1 \leq i < j \leq \tilde{n})$ is the median of the squared Euclidean distances between all pairs of samples in a subset of size $\tilde{n}$ drawn from the data. We choose $\tilde{n}$ as $\min\{n, 1000\}$ to reduce redundant computation. We follow a similar procedure to select the bandwidth for $\mathbf{C}_\zeta$. The second category

---

[1] http://wiki.stat.ucla.edu/socr/index.php/SOCR_Data_MLB_HeightsWeights
[2] https://www.kaggle.com/datasets/budincsevity/szeged-weather

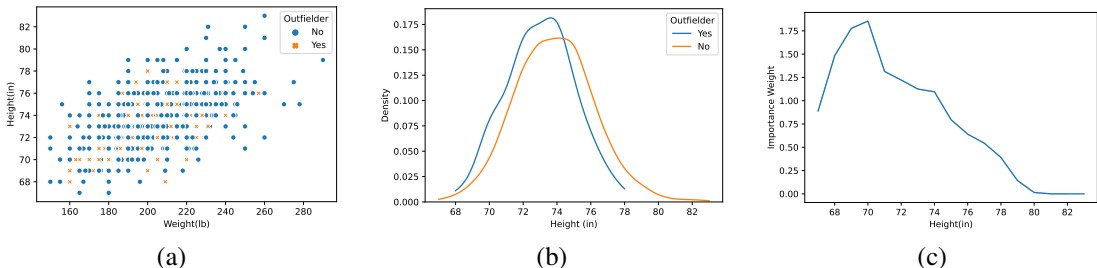

Figure S.1: The SOCR dataset: (a) shows the joint distribution of the players' weights and heights across whether their positions are outfielder; (b) shows the height densities in the training and testing data; (c) shows the empirical estimation of the oracle importance weights.

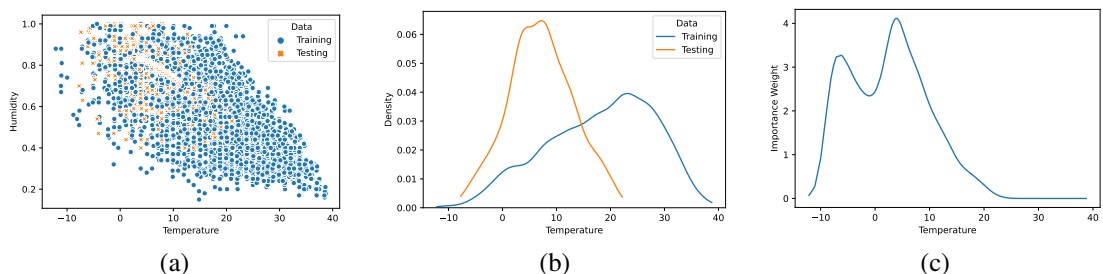

Figure S.2: The Szeged weather dataset: (a) shows the joint distribution of the temperature and humidity; (b) shows the height densities in the training and testing data; (c) shows the empirical estimation of the oracle importance weights.

of hyperparameters is the regularizer parameter $\alpha$. Choosing a large value of $\alpha$ leads to underfitting of the estimator $\rho$ as it is forced to be close to zero. Conversely, a small value of $\alpha$ leads to overfitting of $\rho$. In all of our experiments, we set $\alpha = n^{-1/4}$ to satisfy the condition in Remark 7. Additionally, we adopted the Gaussian kernel in all our experiments.

### C.2 SENSITIVITY ANALYSIS WITH NONLINEAR REGRESSION DATASET

### C.2.1 SIMPLE COMPARISON ON WEIGHT ESTIMATION UNDER DIFFERENT TARGET SHIFT.

In this part, we would like to provide a simple comparison on the weight estimation across different methods. We fixed the sample size $n = 200$ and tune $\mu_t$. The results are shown in Figure S.3. It can be seen that our method keeps the similar pattern as the oracle weight. KMM method has a poor estimation at the boundary. L2IWE method performs worse when $\mu_t$ is large.

### C.2.2 RUNNING TIME COMPARISONS

Under the setting of Section 5.1, we run 50 Monte Carlo experiments to investigate the computational efficiency of the proposed method. We compared the running times of the three methods, ours, KMM and L2IWE, and the results are given in Figure S.4: our method is the fastest among the three methods.

### C.2.3 IMPACT OF THE DENSITY RATIO ESTIMATION

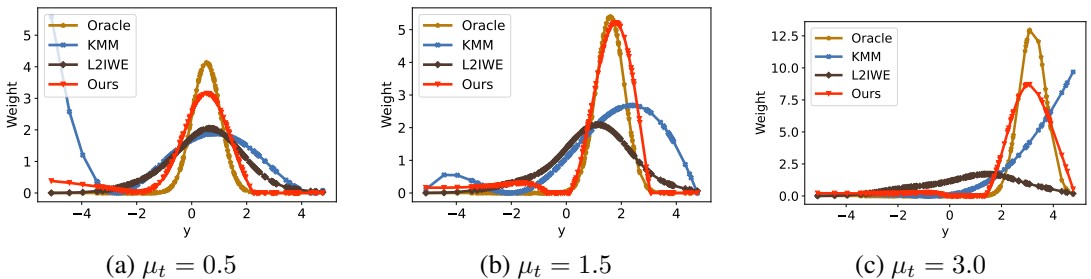

(a) $\mu_t = 0.5$        (b) $\mu_t = 1.5$        (c) $\mu_t = 3.0$

Figure S.3: Weight comparison under $n = 200$ and different $\mu_t$.

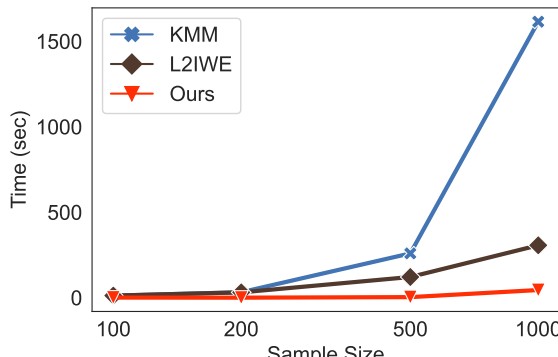

Figure S.4: The total computation time comparison over the 50 replications under different sample sizes.

In this section, we investigate the impact of different density ratio estimation approach on $\eta(\mathbf{x}) = \hat{p}_t(\mathbf{x})/\hat{p}_s(\mathbf{x})$. In our previous experiments, our estimation is to get the direct ratio of two estimated densities $\hat{p}_t(\mathbf{x})$ and $\hat{p}_s(\mathbf{x})$. Instead of this, we can also directly estimate the density ratio. Here we consider the RuLSIF method proposed in Yamada et al. (2013) for the density ratio estimation. Following the experiment design in Section 5.1, we compare the performances under these two methods and show the results in Figure S.5. It can be seen that the two methods performs similar but method with ratio of two estimated densities is better.

### C.2.4 IMPACT OF THE REGULARIZATION PARAMETER $\alpha$

In this part, we conduct the experiments for sensitivity analysis of our proposed ReTaSA method with respect to the regularization parameter $\alpha$. Here all the experiments are reproduced with 50 random trials. For the regularization parameter $\alpha$, it is worth noting that a larger value of $\alpha$ will push the estimated adaptation weight towards 1, while a smaller $\alpha$ will result in a larger estimation variation. In our experiments, we range the value of $\alpha$ from $10^{-2}$ to 1. The results are shown in Figure S.6. Note that in Section C.1 we use the theoretical order $n^{-1/4}$ to tune $\alpha$. So in Figure S.6, we mark the results with theoretical setting with star points. It can be seen that, the performances of our method get better as $\alpha$ increases at the beginning while then get worse with further enlarged $\alpha$. But the theoretical setting can almost reach the best performance status.

### C.3 WEIGHT ESTIMATION ON THE SOCR AND WEATHER DATASETS

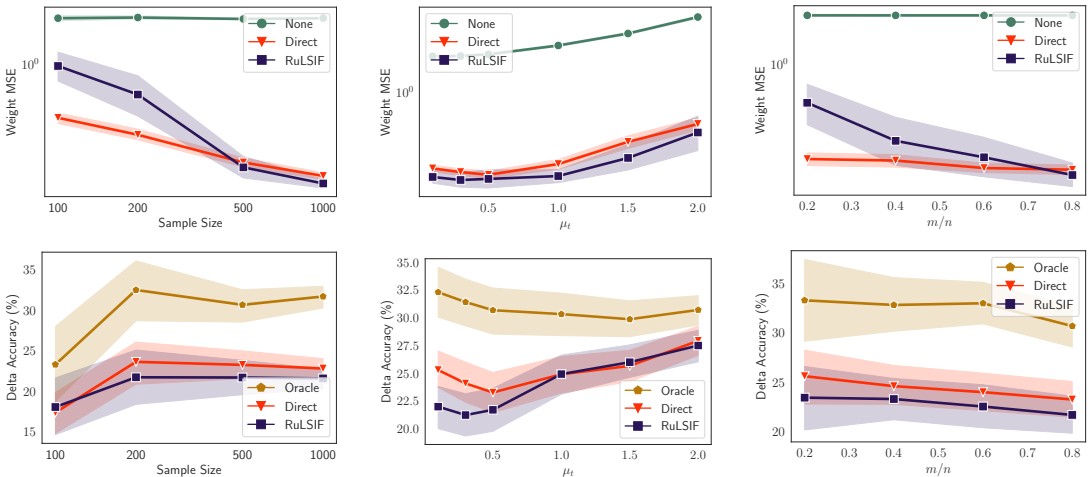

Figure S.5: Comparison of ReTaSA with different density ratio estimation methods: 'Direct' means direct ratio of two estimated density function; 'RuLSIF' is the density ratio estimation method proposed in Yamada et al. (2013). The first row is on the metric of weight MSE and log scale; The second row is on the metric of Delta Accuracy.

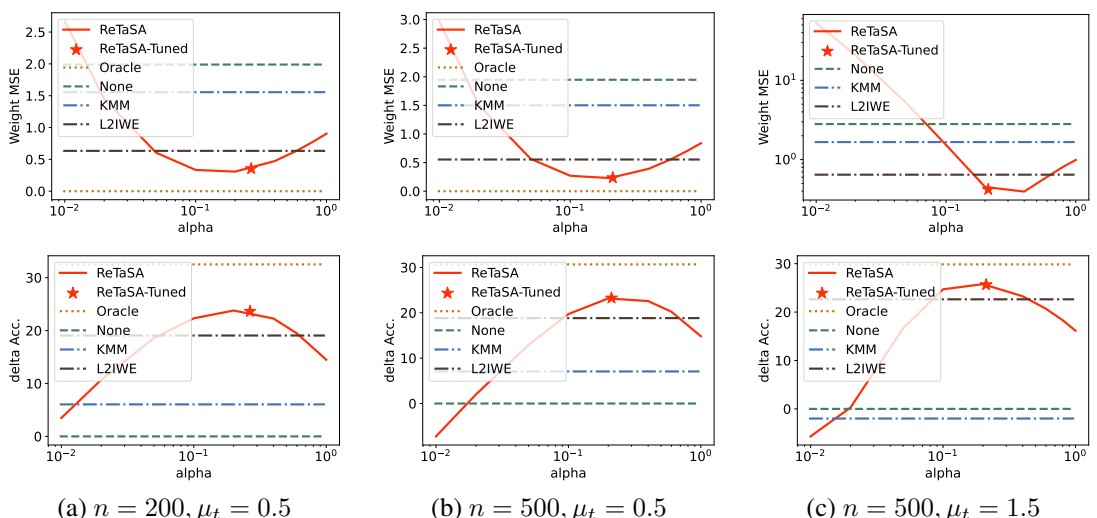

(a) $n = 200, \mu_t = 0.5$      (b) $n = 500, \mu_t = 0.5$      (c) $n = 500, \mu_t = 1.5$

Figure S.6: Sensitivity analysis of regularization parameter $\alpha$ on the synthetic nonlinear regression dataset. The first row is on the metric of weight MSE and the second row is on the metric of $\Delta$Accuracy.

In this part, we'd like to provide a view of the weight estimation on the two real-world datasets in Section 5. For each dataset, we select three trials and show their results in Figure S.7-S.8. It can be seen that the estimated weights from our method are the cloest to the oracle weights. KMM methods has a poor estimation on the boundary and L2IWE method cannot capture the weight pattern well especailly the target shift is intensive.

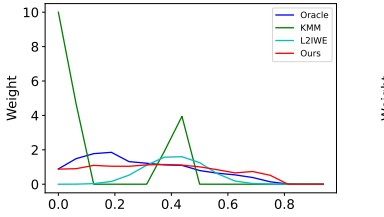 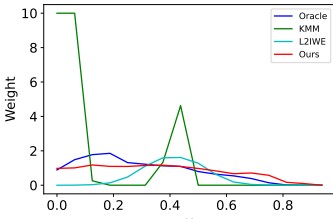 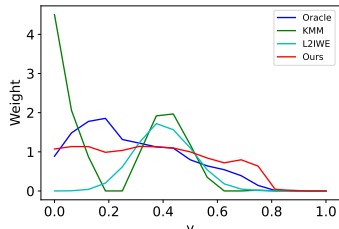

Figure S.7: The weight estimations on the SOCR datasets. The three figures are corresponding to 3 random trials.

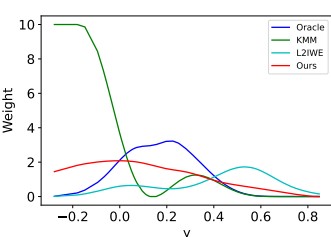 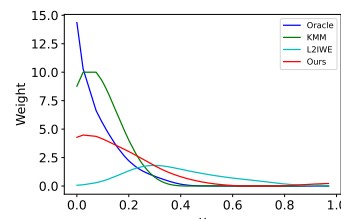 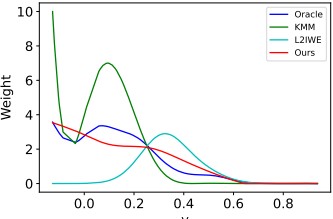

Figure S.8: The weight estimations on the Szeged weather datasets. The three figures are corresponding to first three years.

## C.4 ReTaSA with Black-box Model

The following experiments aim to investigate the performance of ReTaSA combined with the black box model. When the features $\mathbf{x}$ are high-dimensional, the KDE method may suffer from the curse of dimensionality. Using a similar idea from Lipton et al. (2018), we solve the dimensionality problem by finding a "black-box" predictive model, which essentially maps the multivariate $\mathbf{x}$ to a scalar representation $z \equiv \widehat{\mathbb{E}}_s(y|\mathbf{x})$ that $\widehat{\mathbb{E}}_s(y|\mathbf{x})$ is a trained predictive model on the source domain. Note that the target shift assumption holds for $p(z|y)$ (see Lemma 1 in Lipton et al. 2018); thus, we can apply the proposed method by treating $z$ as $\mathbf{x}$ without making any changes.

### C.4.1 Experiment with Multivariate Regression

**Datasets.** Our experiments are on the following two datasets: the synthetic multivariate regression dataset and the UCI Communities and Crime Dataset (Dua & Graff, 2019). In these two datasets, the features $\mathbf{x}$ are multivariate. Without loss of generality, we adopt the linear model as the linear regression model for both black-box mapping and prediction.

1. The synthetic multivariate regression dataset: The data are generated from the model $Y = \mathbf{x}^{\mathrm{T}}\boldsymbol{\beta} + \epsilon$, where $\mathbf{x}, \boldsymbol{\beta} \in \mathbb{R}^5$ and $\epsilon \sim \text{Normal}(0, 1)$. The element values of $\mathbf{x}$ are generated from $\text{Normal}(0, 1)$. We set the regression coefficients with a randomly generated vector $\boldsymbol{\beta} = [1.132, 2.465, 7.776, 0, 0]^{\top}$. The data with top and bottom $5\%$ response values are filtered.

2. The UCI communities and crime dataset: The response variable $y$ is the logarithm of the total number of violent crimes per 100K population (*ViolentCrimesPerPop*), and we predict it with the following features: the number of vacant households (*HousVacant*); the percent of housing occupied (*PctHousOccup*); the percent of households owner occupied (*PctHousOwnOcc*); the percent of vacant housing that is

Table 1: The experimental results of adaptation weight estimation and improved prediction performance on the synthetic multivariate regression datasets. The numbers reported before and after $\pm$ symbolize the mean and standard deviation, respectively. The bold values are the results with the best performance, while Oracle-Adaptation is excluded due to being practically feasible.

| Setting | | Weight MSE | | | $\Delta$Pred. MSE (%) | | |
|---|---|---|---|---|---|---|---|
| $n$ | Shift | None | KMM | Ours | Oracle | KMM | Ours |
| 200 | Mild | $\mathbf{0.01}_{\pm\mathbf{0.00}}$ | $0.10_{\pm0.06}$ | $0.02_{\pm0.01}$ | $0.07_{\pm0.43}$ | $0.07_{\pm1.31}$ | $\mathbf{0.16}_{\pm\mathbf{0.60}}$ |
| | Moderate | $0.07_{\pm0.00}$ | $0.08_{\pm0.05}$ | $\mathbf{0.02}_{\pm\mathbf{0.01}}$ | $0.25_{\pm1.07}$ | $-0.01_{\pm1.46}$ | $\mathbf{0.24}_{\pm\mathbf{0.89}}$ |
| | Intense | $0.31_{\pm0.01}$ | $0.08_{\pm0.05}$ | $\mathbf{0.03}_{\pm\mathbf{0.01}}$ | $0.61_{\pm1.94}$ | $0.68_{\pm1.84}$ | $\mathbf{0.73}_{\pm\mathbf{1.28}}$ |
| 500 | Mild | $\mathbf{0.01}_{\pm\mathbf{0.00}}$ | $0.05_{\pm0.02}$ | $0.01_{\pm0.01}$ | $-0.02_{\pm0.21}$ | $-0.12_{\pm0.61}$ | $-0.03_{\pm0.29}$ |
| | Moderate | $0.07_{\pm0.00}$ | $0.05_{\pm0.03}$ | $\mathbf{0.01}_{\pm\mathbf{0.01}}$ | $0.04_{\pm0.62}$ | $-0.11_{\pm0.81}$ | $\mathbf{0.02}_{\pm\mathbf{0.54}}$ |
| | Intense | $0.32_{\pm0.01}$ | $0.04_{\pm0.03}$ | $\mathbf{0.02}_{\pm\mathbf{0.01}}$ | $1.01_{\pm1.17}$ | $0.86_{\pm1.17}$ | $\mathbf{0.95}_{\pm\mathbf{0.88}}$ |
| 1000 | Mild | $\mathbf{0.01}_{\pm\mathbf{0.00}}$ | $0.02_{\pm0.01}$ | $0.01_{\pm0.01}$ | $0.03_{\pm0.11}$ | $-0.06_{\pm0.21}$ | $-0.02_{\pm0.12}$ |
| | Moderate | $0.08_{\pm0.00}$ | $0.02_{\pm0.01}$ | $\mathbf{0.01}_{\pm\mathbf{0.01}}$ | $0.14_{\pm0.46}$ | $0.08_{\pm0.52}$ | $\mathbf{0.11}_{\pm\mathbf{0.41}}$ |
| | Intense | $0.32_{\pm0.01}$ | $0.03_{\pm0.02}$ | $\mathbf{0.02}_{\pm\mathbf{0.01}}$ | $1.21_{\pm0.75}$ | $\mathbf{1.17}_{\pm\mathbf{0.74}}$ | $1.14_{\pm0.57}$ |

boarded up (*PctVacantBoarded*); the percent of vacant housing that has been vacant more than six months (*PctVacMore6Mos*); the percentage of people 16 and over, in the labor force, and unemployed (*PctUnemployed*); the percentage of people 16 and over who are employed (*PctEmploy*).

**Continuous Target Shift Mechanism.** We propose a bias sampling strategy to change the marginal distribution of the response $y$ to create a target shift. Denote the empirical cumulative distribution function (cdf) of response $y$ in a raw dataset $\mathcal{D}$ as $\widehat{F}(y)$. We first uniformly sample $n$ observations from the raw dataset and treat them as labeled source data. For target data, we generate $m$ random numbers $\{u_i\}_{i=1}^m$ from some distribution with density $g(\cdot)$ on support $[0, 1]$. Then we pick the observations $\{(\mathbf{x}_i, y_i)\}$ from the raw dataset whose quantile of $y_i$ is $u_i$ (i.e., $\sup_y\{y|\widehat{F}(y) \le u_i, y \in \mathcal{D}\}$), for $i = 1, \ldots, m$, as the target data. Next, we show that under such a bias sampling strategy, the importance weight function is given by $\omega(y) = g\left\{\widehat{F}(y)\right\}$: Letting the cdf of the raw data be $F(\cdot)$ and denote $U_1 \sim \text{uniform}(0, 1)$ and $U_2 \sim g(u)$. Then random variable $F^{-1}(U_1)$ has the same distribution as the marignal distribution $p_s(y)$ while $F^{-1}(U_2)$ has the same distribution as $p_t(y)$. Then distribution of $F^{-1}(U_1)$ is given by $p_s(y) = g\{F(y)\} f(y)$, where $f(y)$ is the corresponding pdf of $F(y)$. Similarly, we have $p_t(y) = 1 \cdot f(y)$. Thus, the importance weight function is $\omega(y) = \frac{p_t(y)}{p_s(y)} = g\{F(y)\}$. By replacing $F(\cdot)$ with the empirical $\widehat{F}(\cdot)$, we can obtain the "true" importance weight function.

One can see that if $g(\cdot)$ is uniform, the target and source data are from the same distribution (i.e., $\omega(y) = 1$ for all $y$). In our experiments, we use a truncated normal distribution $g(u) \sim \mathcal{TN}(\mu_t, \sigma_t^2, 0, 1)$. Unless otherwise specified, we set $\mu_t = 0.75$. We use different values of $\sigma_t^2$ to adjust the shift, and a smaller value corresponds to a more severe shift.

**Performance Comparison**. In this part, we showcase the effectiveness of our proposed method in mitigating the negative impact of target shift. Without a loss of generality, the linear regression model is used for the feature dimension reduction and response prediction. We aim to study the methods' performances under different sample sizes and shift settings. In our study, the sample size ranges from $\{200, 500, 1000\}$ and $\sigma_t^2$ is from $\{0.3, 0.5, 0.9\}$. Note that as $\sigma_t^2$ increases, the truncated normal distribution becomes flat, and thus, the shift becomes mild. Thus, we represent the $\sigma_t^2$ settings with three shift levels, {Mild: $\sigma_t^2 = 0.9$, Moderate: $\sigma_t^2 = 0.5$, Intense: $\sigma_t^2 = 0.3$}. We reproduce the experiments with 20 trials for each setting.

Table 2: The experimental results of adaptation weight estimation and improved prediction performance on the UCI communities and crime datasets. The numbers reported before and after $\pm$ symbolize the mean and standard deviation, respectively. The bold values are the results with the best performance, while Oracle-Adaptation is excluded due to being practically feasible.

| Setting | | Weight MSE | | | $\Delta$Pred. MSE (%) | | |
|---|---|---|---|---|---|---|---|
| $n$ | Shift | None | KMM | Ours | Oracle | KMM | Ours |
| 200 | Mild | $\mathbf{0.01}_{\pm\mathbf{0.00}}$ | $1.34_{\pm0.43}$ | $0.01_{\pm0.01}$ | $1.44_{\pm1.87}$ | $-24.35_{\pm29.51}$ | $\mathbf{0.65}_{\pm\mathbf{2.15}}$ |
| | Moderate | $0.08_{\pm0.01}$ | $1.24_{\pm0.47}$ | $\mathbf{0.04}_{\pm\mathbf{0.03}}$ | $8.84_{\pm5.10}$ | $-13.32_{\pm33.03}$ | $\mathbf{4.93}_{\pm\mathbf{3.76}}$ |
| | Intense | $0.33_{\pm0.02}$ | $1.24_{\pm0.61}$ | $\mathbf{0.14}_{\pm\mathbf{0.04}}$ | $31.06_{\pm9.16}$ | $13.72_{\pm34.91}$ | $\mathbf{18.41}_{\pm\mathbf{5.20}}$ |
| 500 | Mild | $\mathbf{0.01}_{\pm\mathbf{0.00}}$ | $1.28_{\pm0.49}$ | $0.01_{\pm0.01}$ | $0.67_{\pm1.45}$ | $-8.64_{\pm10.25}$ | $\mathbf{0.55}_{\pm\mathbf{0.90}}$ |
| | Moderate | $0.08_{\pm0.00}$ | $1.23_{\pm0.54}$ | $\mathbf{0.04}_{\pm\mathbf{0.02}}$ | $7.05_{\pm4.17}$ | $-2.35_{\pm14.38}$ | $\mathbf{4.47}_{\pm\mathbf{2.54}}$ |
| | Intense | $0.33_{\pm0.01}$ | $1.11_{\pm0.59}$ | $\mathbf{0.13}_{\pm\mathbf{0.03}}$ | $29.11_{\pm5.80}$ | $17.22_{\pm14.28}$ | $\mathbf{19.83}_{\pm\mathbf{3.69}}$ |
| 1000 | Mild | $\mathbf{0.01}_{\pm\mathbf{0.00}}$ | $0.92_{\pm0.31}$ | $0.01_{\pm0.01}$ | $0.68_{\pm0.71}$ | $-4.29_{\pm5.23}$ | $\mathbf{0.42}_{\pm\mathbf{0.64}}$ |
| | Moderate | $0.08_{\pm0.00}$ | $0.88_{\pm0.31}$ | $\mathbf{0.03}_{\pm\mathbf{0.01}}$ | $6.45_{\pm2.28}$ | $-0.72_{\pm7.03}$ | $\mathbf{4.52}_{\pm\mathbf{1.54}}$ |
| | Intense | $0.33_{\pm0.01}$ | $0.77_{\pm0.30}$ | $\mathbf{0.10}_{\pm\mathbf{0.02}}$ | $28.51_{\pm3.57}$ | $\mathbf{24.24}_{\pm\mathbf{6.15}}$ | $22.36_{\pm2.41}$ |

The experimental results are summarized in Table 1-2. First, in terms of weight estimation, our method consistently outperforms KMM-Adaptation across all settings. When applied to the crime dataset, KMM-Adaptation experiences significant estimation errors, performing much worse than Non-Adaptation. In contrast, our method maintains low-weight MSE. Secondly, regarding response prediction, our method consistently surpasses KMM-Adaptation, closely approaching the performance of Oracle-Adaptation across a wide range of settings. Even though KMM-Adaptation exhibits slightly better in $\Delta$Prediction MSE than our method with $n = 1000$ and intense shift, it's important to highlight that the corresponding standard deviation of KMM-Adaptation is significantly higher than that of our method. Consequently, we believe that our method consistently outperforms KMM-Adaptation.

### C.4.2 EXPERIMENT WITH MNIST DATASET (POST DISTILLATION)

In this part, we aim to apply our method to the MNIST dataset to evaluate the performance of high-dimensional image datasets. Specifically, our objective is to distill knowledge from a sophisticated teacher model, which has been trained on an extensive labeled dataset, to a smaller student model. The student model aims to attain comparable performance on an unlabeled dataset. This problem can be considered as a sub-task of the model compression and knowledge distillation tasks (Xu et al., 2020; Gou et al., 2021).

In our experiment, we set up the teacher model as a convolutional neural network model with two convolution layers and two fully connected layers. The teacher model is pre-trained with images and their true labels to classify whether the image's digit is odd or even. Then, we use the pre-trained teacher model to evaluate the logits (i.e., $\log(p/(1-p))$ where $p$ is the probability of an image being an odd number) on the unseen images.

The image-logit pairs are used as the dataset for the student model training and evaluation, where the image is the feature, and the logit is the **continuous** response. Our goal is to predict the logit using the image using a student model. The student model is a two-layer multilayer perception; thus, it is much simpler than the teacher model. The responses are the teacher model's logits output so that the problem can be treated as a regression problem.

For the purpose of student model training and evaluation, we split the image-logit datasets according to the images' true labels. Specifically, for an image with an odd number, the probability of assigning it to the source data is $p_{\text{shift}}$, and the probability of being assigned to the target data is $1 - p_{\text{shift}}$. For an image with an even number, the probability of being assigned to the source data is $1 - p_{\text{shift}}$, and the probability of being

assigned to the target data is $p_{\text{shift}}$. If $p_{\text{shift}} = 0.5$, there is no shift between the source and target. If $p_{\text{shift}}$ moves away from $0.5$, the shift is more severe.

We compare KMM, L2IWE and our method under different sample size and shift probability. The experimental results are shown in Figure S.9. It can be seen that our method can consistently achieve good weight estimation. Also the prediction accuracy improvement of our method is much higher than the other two methods.

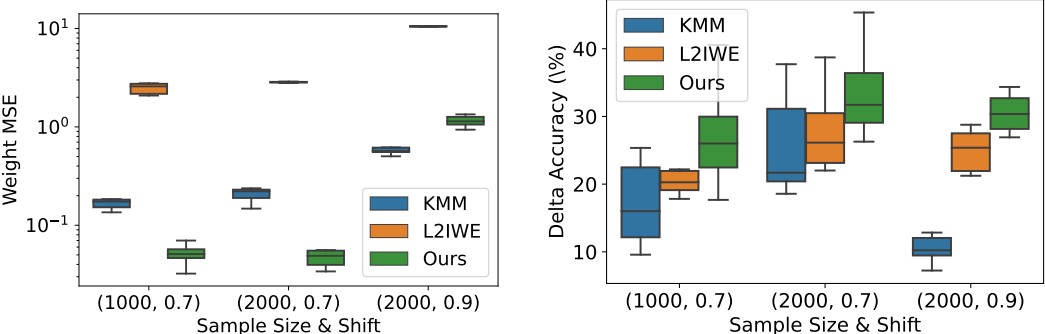

Figure S.9: The performance comparison on MNIST dataset under different sample sizes and shift probability.