# OpenReview forum: "ReTaSA: A Nonparametric Functional Estimation Approach for Addressing Continuous Target Shift"
_ICLR.cc/2024/Conference — ICLR 2024 poster_

### Official Review · Reviewer_SE3s · 2023-10-26

**Soundness:** 3 good
**Presentation:** 3 good
**Contribution:** 3 good
**Rating:** 8
**Confidence:** 3

**Summary:**

This paper studies unsupervised domain adaptation in the context of regression ($\mathbb{R}$-valued labels), under the *label shift* assumption (i.e., that the label distribution $P_Y$ changes while the conditional distribution $P_{X|Y}$ of the features given the labels is invariant. It is first argued that this problem can be solved by training a model under a modified training loss, which is reweighted by the ratio of the label distributions in the test and training domains (the *importance weight function* $\omega$). The remainder of the paper thus focuses on estimating $\omega$. It is shown that $\omega$ satisfies a particular intergral equation, whose other components can be estimated directly from observed data using kernel density estimation. Since this integral equation is typically undercomplete, Tikhonov regularization is added to identify a unique solution. Assuming the relevant data densities are sufficiently regular and bounded, the kernel and bandwidth are carefully selected, etc., the paper provides bounds on the rate at which the estimate of $\omega$ converges to the true $\omega$. The paper then presents experiments on both synthetic and real-world datasets, demonstrating that the proposed approach outperforms a prior kernel-mean-matching approach, both in terms of estimating $\omega$ and out-of-distribution adaptation performance.

**Strengths:**

The motivation and justification for the proposed approach is quite convincing; almost every step seems natural, and so it seems to me like the "right" solution to this problem, under the given assumptions. The high-level writing and flow of the paper are also quite clear. The method is supported both theoretically (with some caveats; see below) and empirically.

**Weaknesses:**

**Major**

1. The paper should discuss the theoretical computational complexity and practical scalability of ReTaSA, in terms of the source and target sample sizes $n$ and $m$, data dimension $p$, etc. Relatedly, under "Related Work", the paper claims "empirical evidence from... our experimental studies confirms that KMM is computationally inefficient in categorical and continuous cases," but I couldn't find evidence of this in the paper.

2. I found several parts of the paper a bit vague or missing some details that would help the reader:
    1. Page 2, last sentence, "the target shift assumption implies that... (1)": I think it would be helpful to include a few more details on the steps by which the target shift assumption implies Eq. (1). I was eventually able to figure this out (using Bayes' rule), but it interrupted my reading of the paper and took a few minutes. This could easily be avoided by adding another intermediate equality in Eq. (1), without increasing the length of the paper.
    2. Page 3, just after Eq. (3), "$T$ and $T^∗$ are adjoint operators because $\langle T\phi, \psi \rangle = \langle \phi, T^* \psi \rangle$": It's not immediately obvious why this is the case. Since this observation is important for the remainder of the paper, please include a more detailed explanation or proof to the main text, or indicate where this could be found (e.g., in an appendix).
    3. Page 3, just after Eq. (4), "where... $\rho(y)$ is a unknown function to be solved": I found this quite confusing because it sounded like $\rho = \omega$. Only later is it explained that $\rho = \omega - 1$. So perhaps this latter fact can be explained a few sentences earlier.

3. First Paragraph of Section 3: $\eta = p_t/p_s$ is estimated as the ratio of two density estimates. There is a significant body of work on density ratio estimation showing that estimating the ratio by the ratio of two density estimates is often suboptimal, both in theory and in practice (see, e.g., [K17, SSK10]). Given this, perhaps the paper should consider using direct density estimation methods for this step, both to improve practical performance and to relax the assumptions (specifically, Assumption 3).

4. There are some gaps between the theoretical results Section 4 and the real-world OOD generalization problem the paper seeks to solve:
    1. Theorem 1 bounds the $L^2$ error of the estimated $\rho$. However, $\rho$ is only a means to re-weighting the risk function to adapt to the test domain (as explained on Page 3), and it's not clear to me whether estimating $\rho$ well in $L^2$ distance is necessary or sufficient to adapt to the test domain. I think the paper should provide some more concrete connection between estimation of $\rho$ in $L^2$ distance and test-domain performance of the new risk minimizer.
    2. It's unclear (to me) how some of the main assumptions in Section 4 relate to the real-world problem being solved; see Major Questions 3. and 4. below.

**Minor**

1. Page 1, Paragraph 2: It's unclear to me why the Sequential Organ Failure Assessment (SOFA) example described here satisfies the label shift assumption (in particular, why $P_{X|Y}$ is invariant between domains). I think a more convincing example here would strengthen the motivation of the paper. Perhaps it is also worth pointing up that label-shift assumptions appear naturally under anti-causal structural assumptions (see, e.g., Section 5 of [S22]).
2. The paper would benefit from some discussion of ReTaSA's limitations or further open questions. Some examples:
    1. The paper focuses on the non-parametric setting. While this makes weak assumptions on the relationship between $x$ and $y$, Theorem 1 suggests that its performance scales poorly with the dimension $p$ of the feature $x$. Perhaps it is worth briefly commenting on whether a parametric variant of ReTaSA (e.g., assuming a linear relationship between $x$ and $y$) could be useful, e.g., for high-dimensional data?
    2. Do the authors believe the rate in Theorem 1 is minimax optimal under Assumptions 1-5?
    3. How robust is ReTaSA to small violations of the label-shift assumption (e.g., small changes in $P_{X|Y}$)?
3. Beginning of Page 4: Tikhonov regularization is added to address non-identifiability (i.e., $T^* T$ might not be invertible). Given this, it might be worth adding a sentence to point out that the regularized criterion has a unique solution (i.e., $\alpha I + T^* T$ is always invertible). This isn't completely obvious, especially in the infinite-dimensional setting.
4. Remark 1: If I am understanding correctly, perhaps it is worth noting that this estimate/approximation is simply the standard Nadaraya-Watson regression estimate of $\mathbb{E}[\rho(y)|x]$.
5. Remark 8, "Therefore, the assumption fits into the regime where the dimension of the feature is smaller than the smoothness level of densities and the order ofthe generalized kernel function.": I didn't understand this sentence. It sounds like it is saying that dim$(x) = p \leq \min\\{k, \ell\\} = \gamma$, but I don't see how this follows from the previous sentence (which is about $\alpha$).
6. Page 7, under "Evaluation Metrics", "We conducted all experiments with 50 replications on a Mac-Book Pro equipped with a 2.9 GHz Dual-Core Intel Core I5 processor and 8GB of memory.": This seems like the wrong place to include this information. Perhaps it should be in the first paragraph of Section 5?
7. Page 7, under "Experimental Results", Typo: "performs significantly better KMM-Adaptation" should be "performs significantly better *than* KMM-Adaptation"
8. Figure 2: The lines plotted here are essentially all flat, so the plot does not illustrate much. Perhaps it would be useful to show a larger range of (smaller) sample sizes?
9. Figure 3: I think the sub-captions are incorrect (they both say "vs Sample Size" but the $x$-axis here is $\mu_t$).
10. All Figures: Please increase the font size of the text in the plots (axis labels, legends, etc.).

**References**

[K17] Kpotufe, S. (2017, April). Lipschitz density-ratios, structured data, and data-driven tuning. In Artificial Intelligence and Statistics (pp. 1320-1328). PMLR.

[S22] Schölkopf, B. (2022). Causality for machine learning. In Probabilistic and Causal Inference: The Works of Judea Pearl (pp. 765-804).

[SSK10] Sugiyama, M., Suzuki, T., & Kanamori, T. (2010). Density ratio estimation: A comprehensive review (statistical experiment and its related topics). 数理解析研究所講究録, 1703, 10-31.

**Questions:**

**Major**

1. I found Definition 2 quite confusing, for a few reasons:
    1. I don't see where Definition 2 is used anywhere in the paper.
    2. In contrast to Section 3, where the (presummably translation invariant?) kernels are written as a univariate function, the kernel here is written as a bivariate function. Is there a reason for this?
    3. I don't understand the condition $k_h(x, y) = 0$ if $x \notin [y - 1, y] \cap \mathcal{C}$. For example, usually, bivaraite kernels are symmetric in their arguments, but this appears not to be the case here.
    4. The use of $x$ and $y$ is a bit confusing here, as it suggests the kernel is applied to the covariate $x$ and the label $y$ discussed earlier in the paper (but I don't think this is the intent, since, e.g., the condition $x \notin [y - 1, y]$ really would not make any sense in this case). Perhaps different variables (e.g., $z_1$ and $z_2$) should be used here?
2. Page 3, Eqs. (2)-(3): I don't understand why $T$ and $T^*$ map $L^2$ to $L^2$. Is this an additional implicit assumption? Or does it follow from the forms (conditional expectations) of $T$ and $T^*$? Relatedly, just after Eq. (4), "$\mathbb{E}_s\\{\eta(x)\\} = 0$ so that $\eta(x) \in L^2(x)$": I didn't understand this implication; why is $\mathbb{E}_s\\{\eta^2(x)\\} < \infty$?
3. Assumption 2: Although I'm familiar with math behind this assumption, I found it hard to understand its ramifications in this context. I understand that the $\beta$-regularity space is the set of $\rho$ such that $T^* T \rho$ is approximately invertible, but is there a more concrete intuition, or maybe some examples, of what $\beta$-regularity looks like when $T$ is the conditional expectation operator? For example, if if $\\{\phi_i\\}_{i = 1}^\infty$ is the Fourier basis, then $\beta$ is related to the smoothness/differentiability of $\rho$.
4. Assumption 5: Similar to the previous point, can you provide some intuition or examples for when Assumption 5 is satisfied?


**Minor**

1. Assumption 4: "Letting $\gamma = \min\\{k, \ell\\}$...": $\gamma$ is never used again in this assumption. Is this an error? Perhaps $\gamma$ should be defined in Assumption 5 instead?
2. Remark 6: Does "consistent" here mean "in operator norm"? If so, please make this more explicit.
3. I didn't understand the definition of "Delta Accuracy" in the experiments. Could you please provide a mathematical formula? Would $100\%$ correspond to perfect prediction?
4. Page 8, Last Sentence, "In each trial, we... randomly select 80% of the players with the other positions as the training source data.": If I understand correctly, the idea here is to use the bootstrap to obtain confidence intervals on performance. If so, this resampling should be done *with replacement*. Or is there a different reasoning here?
5. Page 9, First Paragraph, "For the temperature value shift between the training source and testing dataset, we use the humidity for the importance weight estimation.": I didn't understand this sentence. What does it mean to "use the humidity for the importance weight estimation" (as opposed to using all of the available features)?
6. Page 9, Second Paragraph, "our method improves about 4% for the SOCR dataset": I don't see this 4% in Table 1. What exactly is this refering to?

---

> ### Author Response · Authors · 2023-11-16
> **Responses to Reviewer SE3s**
>
> ## **Weaknesses:**
>
> ### **Major**
>
> > 1. *The paper should discuss the theoretical computational complexity and practical scalability of ReTaSA, in terms of the source and target sample sizes $n$ and $m$, data dimension $p$, etc. Relatedly, under "Related Work", the paper claims "empirical evidence from ... our experimental studies confirms that KMM is computationally inefficient in categorical and continuous cases," but I couldn't find evidence of this in the paper.*
>
>  **Our Response**: Thanks for pointing this out. Our method computationally faster than the KMM method is because our method only needs to solve a linear equation, while KMM needs to optimize the constraint quadratic programming problem. More specifically, the major computational cost in our method is inverting a $n\times n$ matrix, whose computational complexity is generally $\mathcal{O}(n^3)$. The KMM method proposed by Zhang et al. (2013) needs to optimize a quadratically constrained ***quadratic*** program with a $n\times n$ matrix.  We believe the computational efficiency can be attributed to the fact that our proposed method only requires a one-step linear system solving while the KMM method (as well as the added L2IWE method) needs to loop through iteration in the optimization process. **From the empirical aspect, we detail the running time comparisons in Figure S.4 in Section C.2.2 of the supplementary materials**.
>
> > 2. *I found several parts of the paper a bit vague or missing some details that would help the reader:*
> > 2.1. *Page 2, last sentence, "the target shift assumption implies that... (1)": I think it would be helpful to include a few more details on the steps by which the target shift assumption implies Eq. (1). I was eventually able to figure this out (using Bayes\' rule), but it interrupted my reading of the paper and took a few minutes. This could easily be avoided by adding another intermediate equality in Eq. (1) without increasing the length of the paper.*
>
> **Our Response**: Thanks for the suggestion. We have provided more details in the derivation in the revision in Equation (1).
>
> > 2.2. *Page 3, just after Eq. (3), "$T$ and $T^{\ast}$ are adjoint operators because $\langle T\phi,\psi\rangle = \langle\phi, T^{\ast}\psi\rangle$\": It\'s not immediately obvious why this is the case. Since this observation is important for the remainder of the paper, please include a more detailed explanation or proof in the main text or indicate where this could be found (e.g., in an appendix).*
>
> **Our Response**: We have refined the definition of inner products and added a few more lines of equations between equation (3) and (4) to validate this fact.
>
> > 3. *Page 3, just after Eq. (4), "where... $\rho(y)$ is an unknown function to be solved\": I found this quite confusing because it sounded like $\rho = \omega$. Only later is it explained that $\rho = \omega - 1$. So perhaps this latter fact can be explained a few sentences earlier.*
>
> **Our Response**: Thanks for the suggestion; we have clarified the usage of notation $\rho$ as you suggested and edited the identifiability subsection to reflect this change.
>
> > 3. *First Paragraph of Section 3: $\eta = p_{t}/p_{s}$ is estimated as the ratio of two density estimates. There is a significant body of work on density ratio estimation showing that estimating the ratio by the ratio of two density estimates is often suboptimal, both in theory and in practice (see, e.g., \[K17, SSK10\]). Given this, perhaps the paper should consider using direct density estimation methods for this step, both to improve practical performance and to relax the assumptions (specifically, Assumption 3).*
>
> **Our Response**: Thanks for providing closely connected references. Utilizing direct density estimation to yield an improved rate of consistency will be an interesting future theoretical work to pursue. In the meantime, we believe that assumption 3 in our setting is natural as the equation (1) would not be valid if violated. We have added a remark to explain connections between the assumption 3 and the equation (1).
>
> Furthermore, in the revised paper, we compared two ways of computing the density ratio: directly computing the ratio of two estimated densities and the Relative unconstrained Least-Squares Importance Fitting (RuLSIF) method (Yamada et al. (2011)), which is proposed by the author of [SSK10]. The numerical study shows that these two methods have similar performance. The details of the numerical comparison can be found in Section C.2.3 of the Supplementary Materials.

---

> ### Author Response · Authors · 2023-11-16
>
> > 4. *There are some gaps between the theoretical results of Section 4 and the real-world OOD generalization problem the paper seeks to solve:*
> > 4.1. *Theorem 1 bounds the $L^{2}$ error of the estimated $\rho$. However, $\rho$ is only a means to re-weighting the risk function to adapt to the test domain (as explained on Page 3), and it\'s not clear to me whether estimating $\rho$ well in $L^{2}$ distance is necessary or sufficient to adapt to the test domain. I think the paper should provide some more concrete connection between estimation of $\rho$ in $L^{2}$ distance and test-domain performance of the new risk minimizer.*
>
> **Our Response**: Thanks for pointing this out. The key reason for the weight estimation is motivated by the loss relationship between training and testing datasets.
> $$
> \mathbb{E}_t[\ell\{f(x),y\}]=\mathbb{E}_s[\omega(y)\ell\{f(x),y\}]
> $$
> With a well-estimated $\rho$, we can recover the importance weight function $\omega$ accurately, which can facilitate us to learn a predictive model with respect to the test-domain distribution. As $\rho$ is a function, we use the $L^2$-distance to measure the estimation accuracy in the functional space.
>
> > 4.2. *It\'s unclear (to me) how some of the main assumptions in Section 4 relate to the real-world problem being solved; see Major Questions 3. and 4. below.*
>
> **Our Response**: Thanks for the comment. Please see our responses below under Major Questions 3-4.

---

> > ### Author Response · Authors · 2023-11-16
> >
> > ### **Minor**
> >
> > > 1. *Page 1, Paragraph 2: It\'s unclear to me why the Sequential Organ Failure Assessment (SOFA) example described here satisfies the label shift assumption (in particular, why $P_{X|Y}$ is invariant between domains). I think a more convincing example here would strengthen the motivation of the paper. Perhaps it is also worth pointing up that label-shift assumptions appear naturally under anti-causal structural assumptions (see, e.g., Section 5 of \[S22\]).*
> >
> > **Our Response**: Thanks for the suggestion. First, we want to clarify the relationship between label shift and biased sampling*.* Suppose $r$ is a binary random variable indicating a patient is sent to a big or small hospital (i.e., domain), and $r$ only depends on the SOFA score (i.e., the severity of the condition): $p(r|x,y)=p(r|y)$, which is equivalent to $r$ and $x$ are independent conditional on $y$. This conditional independence relationship can be rewritten as $p(x|y,r)=p(x|y)$. In other words, the conditional distribution of $p(x|y)$ is domain invariant (in this case, $y$ does not have to cause $x$), thus honoring the target shift assumption. The anti-causal setting is also an important motivation for target shift, and we have discussed it in the revision of the paper.
> >
> > > 2. *The paper would benefit from some discussion of ReTaSA\'s limitations or further open questions. Some examples:*
> > > 2.1. *The paper focuses on the non-parametric setting. While this makes weak assumptions on the relationship between $x$ and $y$, Theorem 1 suggests that its performance scales poorly with the dimension $p$ of the feature $x$. Perhaps it is worth briefly commenting on whether a parametric variant of ReTaSA (e.g., assuming a linear relationship between $x$ and $y$) could be useful, e.g., for high-dimensional data?*
> >
> > **Our Response**: Indeed, nonparametric methods often suffer from the curse of dimensionality. To address this issue, our suggestion is to adopt the black box shift estimation (BBSE) method proposed by Lipton et al. (2018), where the high-dimensional features $x$ are mapped to $\widehat{E}_s(y|x)$, where $\widehat{E}_s(\cdot|x)$ is a fitted predictive model. In this revision, we have adopted this approach and added numerical studies on data with many covariates. As you have suggested, it is possible to solve this problem by assuming a parametric model for the importance weight function. One of the simplest models is a polynomial model. For example, $\omega(y)=\exp(\alpha+\beta_1 y+\beta_2 y^2)$. Then, we can estimate the parameters using a semiparametric framework (we do not have to specify a parametric model for $p_s(y|x)$). Similar work has been done for the classification task (e.g., Tian et al., 2023). We want to comment that although we can address the high-dimensional using either the black-box or semiparametric methods, there is no free lunch. If we adopt the black-box method, we need a stronger identifiability assumption. After the mapping, the mapped $\widehat{E}_s(y|x)$ needs to satisfy the completeness condition, which is stronger than the current identifiability assumption. On the other hand, adapting a parametric model for the density ratio will impose model constraints and introduce model misspecification errors.
> >
> > > 2.2. *Do the authors believe the rate in Theorem 1 is minimax optimal under Assumptions 1-5?*
> >
> > **Our Response**: Thanks for pointing out this issue. We admit that our theoretical result does not follow the minimax optimal framework. The goal of our Theorem 1 is to demonstrate the consistency of the estimated importance weight function and provide the theoretical variation. To reach the minimax optimal, it will involve more assumptions and analysis for this non-parametric approach. We would like to put it for the future investigation.
> >
> > > 2.3. *How robust is ReTaSA to small violations of the label-shift assumption (e.g., small changes in $P_{X|Y}$)?*
> >
> > **Our Response:** Our proposed method relies on the target (label) shift assumption. Acknowledging the practical reality of open-world scenarios, it is inevitable that this assumption may be violated. In line with prior research, such as Sahoo et al. 2022, a common strategy to quantify these minor violations involves imposing a divergence-based constraint on $p(x|y)$. Specifically, it is posited that there exists a $\epsilon>0$ and a distance (or divergence) measure, denoted as $D$, satisfying $D(p_s(x|y), p_t(x|y))\leq\epsilon$. Distances encompass well-known metrics like $f$-divergences and Wasserstein distance. Recognizing the potential for a more comprehensive framework to characterize robustness, as outlined above, we appreciate this valuable suggestion and intend to explore it further in future research endeavors.

---

> > > ### Author Response · Authors · 2023-11-16
> > >
> > > > 3. *Beginning of Page 4: Tikhonov regularization is added to address non-identifiability (i.e., $T^{\ast}T$ might not be invertible). Given this, it might be worth adding a sentence to point out that the regularized criterion has a unique solution (i.e., $\alpha I + T^{\ast}T$ is always invertible). This isn't completely obvious, especially in the infinite-dimensional setting.*
> > >
> > > **Our Response**: Thanks for the comment; we have emphasized the uniqueness of the solution to the first-order equation in the revision.
> > >
> > > > 4. *Remark 1: If I am understanding correctly, perhaps it is worth noting that this estimate/approximation is simply the standard Nadaraya-Watson regression estimate of $\mathbb{E}\lbrack\rho(y)|x\rbrack$.*
> > >
> > > **Our Response**: Thanks for the comment. Yes, the estimate can be viewed as the standard Nadaraya-Watson regression estimates, and we have modified our presentation accordingly.
> > >
> > > > 5. *Remark 8, \"Therefore, the assumption fits into the regime where the dimension of the feature is smaller than the smoothness level of densities and the order of the generalized kernel function.\": I didn\'t understand this sentence. It sounds like it is saying that dim$(x) = p \leq \min\{ k,\ell\} = \gamma$, but I don't see how this follows from the previous sentence (which is about $\alpha$).*
> > >
> > > **Our Response**: Thanks for the comment, and sorry for the confusion. We rephrased the original remark 8 to the current remark 5 & 7. In remark 5, we discuss the relationship between dimension $p$ and the sample sizes $m$ and $n$. In remark 7, we discuss the relationship between dimension $p$ and regularization parameter $\alpha$, which also involves the smoothness of the generalized kernel functions.
> > >
> > > > 6. *Page 7, under \"Evaluation Metrics\", \"We conducted all experiments with 50 replications on a Mac-Book Pro equipped with a 2.9 GHz Dual-Core Intel Core I5 processor and 8GB of memory.\": This seems like the wrong place to include this information. Perhaps it should be in the first paragraph of Section 5? Page 7, under \"Experimental Results\", Typo: \"performs significantly better KMM-Adaptation\" should be \"performs significantly better than KMM-Adaptation\"*
> > >
> > > **Our Response**: Thanks for pointing this out. We have rephrased the presentation and fixed the typo.
> > >
> > > > 7. *Figure 2: The lines plotted here are essentially all flat, so the plot does not illustrate much. Perhaps it would be useful to show a larger range of (smaller) sample sizes?*
> > >
> > > **Our Response**: Thanks for raising this issue. Here, we want to clarify that the lines look flat due to the consistently significant improvement of our method compared with KMM-Adaptation and None-Adaptation methods under different sample sizes. Specifically, for our method, we list the weight MSE in the following table: MSE is about 0.45 at sample size = 100 and less than 0.2% at sample size = 1000. Thus, the improvement is more than 50% by increasing the sample size from 100 to 1000 under our method.
> > >
> > > | Sample Size | Weight MSE |
> > > | --- | --- |
> > > | 100 | 0.454114 |
> > > | 200 | 0.353673 |
> > > | 500 | 0.235435 |
> > > | 1000 | 0.192919 |
> > >
> > > > 8. *Figure 3: I think the sub-captions are incorrect (they both say \"vs Sample Size\" but the $x$-axis here is $\mu_{t}$).*
> > >
> > > **Our Response**: Thanks for pointing out the typo. We have fixed it.
> > >
> > > > 9. *All Figures: Please increase the font size of the text in the plots (axis labels, legends, etc.).*
> > >
> > > **Our Response**: Thanks for the suggestion. We have iterated all the figures in the main paper with a larger font size.

---

> > > > ### Author Response · Authors · 2023-11-16
> > > >
> > > > ## **Questions:**
> > > >
> > > > ### **Major**
> > > >
> > > > > 1. *I found Definition 2 quite confusing, for a few reasons:*
> > > > > 1.1 *I don\'t see where Definition 2 is used anywhere in the paper.*
> > > > > 1.2. *In contrast to Section 3, where the (presummably translation invariant?) kernels are written as a univariate function, the kernel here is written as a bivariate function. Is there a reason for this?*
> > > > > 1.3. *I don\'t understand the condition $k_{h}(x,y) = 0$ if $x \notin \lbrack y - 1,y\rbrack \cap \mathcal{C}$. For example, usually, bivaraite kernels are symmetric in their arguments, but this appears not to be the case here.*
> > > > > 1.4. *The use of $x$ and $y$ is a bit confusing here, as it suggests the kernel is applied to the covariate $x$ and the label $y$ discussed earlier in the paper (but I don\'t think this is the intent, since, e.g., the condition $x \notin \lbrack y - 1,y\rbrack$ really would not make any sense in this case). Perhaps different variables (e.g., $z_{1}$ and $z_{2}$) should be used here?*
> > > >
> > > > **Our Response**: **a)**. We have moved Definition 2, which was originally reserved for generalized kernel function, to Section 3, as it has close relevance to the methodology section. **b)**. Thank you for pointing it out; we have corrected notations for the kernel function consistently throughout the paper. **c)** Thanks for the comment, and sorry for the confusion. Without loss of the generality, in the theoretical analysis, we assume that the data are standardized in a unit ball so that the mentioned condition is satisfied. We have used the generalized kernel function, which does not necessarily include symmetric kernels. We have fixed the kernel notations to be consistent throughout the paper. We refer to (Muller.1991) for more details on the generalized kernel. **d).** Thank you for the suggestion. We have modified the notations to avoid confusion.
> > > >
> > > > > 2. *Page 3, Eqs. (2)-(3): I don\'t understand why $T$ and $T^{\ast}$ map $L^{2}$ to $L^{2}$. Is this an additional implicit assumption? Or does it follow from the forms (conditional expectations) of $T$ and $T^{\ast}$? Relatedly, just after Eq. (4),
> > > > \"$\mathbb{E}{s}\{\eta(x)\} = 0$ so the at $\eta(x) \in L^{2}(x)$\": I didn\'t understand this implication; why is $\mathbb{E}{s}\{\eta^{2}(x)\} < \infty$?*
> > > >
> > > > **Our Response**: Thanks for the comment. First, this is not an additional assumption but a definition. In Eqs (2)-(3), we want to present that *$T$ and $T^{\ast}$ are two conditional expectation operators to map the functions between $L^{2}(x)$ and $L^{2}(y)$.* We believe the confusion arose from the definition of $L^2$ space we use. We have clarified the definition of $L^2$ space on page 3. Secondly, in our analysis, *we only consider the mean-zero and square-integrable function space. And $\mathbb{E}_{s}\{\eta^{2}(x)\} < \infty$ is due to the definition of square-integrable.*

---

> > > > > ### Author Response · Authors · 2023-11-16
> > > > >
> > > > > > 3. *Assumption 2: Although I\'m familiar with math behind this assumption, I found it hard to understand its ramifications in this context. I understand that the $\beta$-regularity space is the set of $\rho$ such that $T^{\ast}T\rho$ is approximately invertible, but is there a more concrete intuition, or maybe some examples, of what $\beta$-regularity looks like when $T$ is the conditional expectation operator? For example, if $\{\phi_{i}\}_{i = 1}^{\infty}$ is the Fourier basis, then $\beta$ is related to the smoothness/differentiability of $\rho$.*
> > > > >
> > > > > **Our Response**: The $\beta$-regularity space amounts to a condition of the relationship between the **smoothness** on the function $\rho(y)$ (i.e., rate of decay of its Fourier coefficients) and the **degree of ill-posedness** on the operator $T$ (i.e., rate of decay of its singular values). For example, suppose the operator is severely ill-posed (i.e., the rate of decay of the singular values of $T$ is fast), the condition says that we will also need the function $\rho(y)$ to be very smooth (rapidly decay). Otherwise, $\rho(y)$ is no longer in the $\beta$-regularity space.
> > > > >
> > > > > In addition to the solution's existence (as you have mentioned: invertible), it is crucial for the Moore Penrose generalized inverse of $T$ to be **continuous**. If it is not continuous, the solution is unstable (i.e., explosive behavior). However, when we generalize a functional estimation from a parametric setting to a nonparametric one (i.e., a limit of finite dimensional problems), the Moore-Penrose generalized inverse is not continuous. Thus, there is no way we can obtain a consistent estimator because we do not have the true function $\eta$ and have to estimate it (where the estimation will introduce “perturbation” (i.e., estimation error) in the input $\eta$). After recasting the problem from $L^2(X)$ and $L^2(Y)$ to the $\beta$-regularity spaces. The ill-posed problem becomes well-posed with topologies on the $\beta$-regularity spaces. In other words, the Moore-Penrose generalized inverse is **bounded** under the new topologies, thus leading to a continuous solution. Thus making it possible to have a consistent estimation of the function $\rho(y)$.
> > > > >
> > > > > > 4. *Assumption 5: Similar to the previous point, can you provide some intuition or examples for when Assumption 5 is satisfied?*
> > > > >
> > > > > **Our Response**: Thanks for the comment we claim the original assumption 5 can be justified under all other previous assumptions and thus presented as a lemma. We have removed this assumption from the main paper and prove it in the supplementary.

---

> > > > > > ### Author Response · Authors · 2023-11-16
> > > > > >
> > > > > > ### **Minor**
> > > > > >
> > > > > > > 1. *Assumption 4: \"Letting $\gamma = \min( k,\ell)$\...\": $\gamma$ is never used again in this assumption. Is this an error? Perhaps $\gamma$ should be defined in Assumption 5 instead?*
> > > > > >
> > > > > > **Our Response**: Thanks for the catch; this condition is not necessary for assumption 4, so we have deleted it.
> > > > > >
> > > > > > > 2. *Remark 6: Does \"consistent\" here mean \"in operator norm\"? If so, please make this more explicit.*
> > > > > >
> > > > > > **Our Response**: Yes, the consistency is under the operator norm. We have added an additional phrase in remark 5, which was originally remark 6, to clarify the norm. In the meantime, one can also show that it is consistent under the Hilbert-Schmidt norm. Furthermore, we have moved this assumption to the appendix and provided references for its justification.
> > > > > >
> > > > > > > 3. *I didn\'t understand the definition of \"Delta Accuracy\" in the experiments. Could you please provide a mathematical formula? Would $100$ correspond to perfect prediction?*
> > > > > >
> > > > > > **Our Response**: Sorry for the confusion. Here, the Delta Accuracy is defined as the percentage of improved prediction MSE compared with Non-Adaptation. Mathematically, it can be formulated as \[$\Delta$ Accuracy = (prediction MSE of None Adaptation- prediction MSE of proposed method ) /  prediction MSE of None Adaptation * 100\%\]. The larger value represents better prediction performance while 100% indeed means the perfect prediction because the prediction MSE of the proposed method is 0 then.
> > > > > >
> > > > > > > 4. *Page 8, Last Sentence, \"In each trial, we\... randomly select 80% of the players with the other positions as the training source data.\": If I understand correctly, the idea here is to use the bootstrap to obtain confidence intervals on performance. If so, this resampling should be done with replacement. Or is there a different reasoning here?*
> > > > > >
> > > > > > **Our Response**: Yes, we used bootstrap to obtain confidence intervals for performances.  Specifically, in each trial, we random sample 80% training data so that there is no duplicated training data in the sample trial. While across trials, the random sampling is with replacement. This bootstrap method is also valid in the literature, e.g., Sitter et al. (1992). When bootstrapping with replacement, the bootstrap sample size is the same as the original data; for bootstrapping without replacement, the bootstrap sample size is smaller than the original one (in our case, 80%). But under the independent and identically distributed (i.i.d.) assumption, both methods are valid because, essentially, both methods sample from the same empirical distribution.
> > > > > >
> > > > > > > 5. *Page 9, First Paragraph, \"For the temperature value shift between the training source and testing dataset, we use the humidity for the importance weight estimation.\": I didn\'t understand this sentence. What does it mean to \"use the humidity for the importance weight estimation\" (as opposed to using all of the available features)?*
> > > > > >
> > > > > > **Our Response**:  Sorry for the confusion. We want to clarify that we only select the *humidity for the weight estimation. This is* because, from Meteorology study, it has been widely known that there is a causality relationship between humidity and temperature: the high temperature will lead to low humidity, which is also validated in our exploratory data analysis in the supplementary Section C.1. *However, the other features may not satisfy the target shift assumption.*
> > > > > >
> > > > > > > 6. *Page 9, Second Paragraph, \"our method improves about 4% for the SOCR dataset\": I don\'t see this 4% in Table 1. What exactly is this referring to?*
> > > > > >
> > > > > > **Our Response**: Sorry for the confusion. Here is how the 4% comes: The KMM method has the $\Delta$ Prediction MSE as 3%, and our method has it as 6.7%. Thus, the difference is about 6.7% -3% = 3.7%, which we mean by about 4%.
> > > > > >
> > > > > >
> > > > > > ## References
> > > > > >
> > > > > > Hans-Georg Muller. Smooth Optimum Kernel Estimators Near Endpoints, Biometrika, Vol.78, No.3. (1991)
> > > > > >
> > > > > > Sahoo, R., Lei, L., & Wager, S. (2022). Learning from a biased sample. *arXiv preprint arXiv:2209.01754*.
> > > > > >
> > > > > > Sitter, Randy Rudolf. "Comparing three bootstrap methods for survey data." *Canadian Journal of Statistics* 20.2 (1992): 135-154.
> > > > > >
> > > > > > Tian, Q. Zhang, X., Zhao, J., (2023). ELSA: Efficient Label Shift Adaptation through the Lens of Semiparametric Models. ICML

---

> > > > > > ### Comment · Reviewer_SE3s · 2023-11-16
> > > > > > **Clarifying my question about $\beta$-regularity**
> > > > > >
> > > > > > Thanks to the authors for their detailed responses.
> > > > > >
> > > > > > Regarding my Major Question 3. about Assumption 2.: Sorry, I realize my question was a bit too vague earlier. I was really asking about some intuition regarding what the spectrum *of a conditional expectation operator* looks like. Is there any nice way to think about this in terms of the underlying joint distribution $p(x, y)$ (if not in general, at least in a simple example)? It's ok if the answer is negative, but it would be great if there's some way to understand this $\beta$-regularity assumption in the context of the original problem of regressing y over x under target shift. At present, I have no idea whether this assumption is reasonable in a realistic problem.

---

> ### Comment · Reviewer_SE3s · 2023-11-16
> **Clarifying the "gap" between the $L^2$ error and test-domain performance**
>
> Thanks to the authors for responding to my comments. For my comment 4.2, I was hoping for was some more detailed discussion along the following lines.
>
> Ultimately, rather than estimating $\omega$, what we care about is something like the test-domain generalization error
> $$E_s[\omega(y) \ell(\hat f(x), y))] - \frac{1}{n} \sum_{i = 1}^n \hat\omega(y_i) \ell(\hat f(x_i), y_i)$$
> of the weighted risk minimization
> $$\hat f = \arg\min_f \frac{1}{n} \sum_{i = 1}^n \hat\omega(y_i) \ell(f(x_i), y_i).$$
> One approach to bound this generalization error is by bounding the difference between the true and empirical risks uniformly over the hypothesis class, i.e.,
> $$\sup_f \left| \frac{1}{n} \sum_{i = 1}^n \hat\omega(y_i) \ell(f(x_i), y_i) - \mathbb{E}_s[\omega(y) \ell(f(x), y))] \right|.$$
>
> For simplicity, let's assume the source sample size $n$ is huge, the hypothesis class is reasonably bounded, and $y$ is bounded almost surely, so that the true and empirical risk functionals are basically identical; i.e., $\sup_f \left| \frac{1}{n} \sum_{i = 1}^n \hat\omega(y_i) \ell(f(x_i), y_i) - \mathbb{E}_s[\hat\omega(y) \ell(f(x), y)] \right| \approx 0$ (this can be bounded more formally by standard techniques). Then, the generalization error mostly comes from the error in estimating $\omega$, namely
> $$\sup_f \left| \mathbb{E}_s[\omega(y) \ell(f(x), y)] - \mathbb{E}_s[\hat \omega(y) \ell(f(x), y)] \right|
>   = \sup_f \mathbb{E}_s[(\omega(y) - \hat\omega(y)) \ell(f(x), y)]
>   \leq \sqrt{\mathbb{E}_s[(\omega(y) - \hat\omega(y))^2] \sup_f \mathbb{E}_s[\ell^2(f(x), y)]}.$$
> Focusing on the term involving $\hat\omega$, it looks like we care about the *weighted* $L^2$ (weighted by $p_s(y)$) error, rather than the unweighted $L^2$ error that is bounded in Theorem 1. However, the weighted $L^2$ error can be bounded in terms of the unweighted $L^2$ error if we additionally assume $p_s(y)$ is bounded above.
>
> This gives one bound on the test-domain generalization error in terms of the $L^2$ bound in Theorem 1, but there may be better ways (e.g., tighter bounds or making weaker assumptions). So I asking the authors to think a bit about this gap and to justify that bounding the $L^2$ error in Theorem 1 is useful and/or necessary.

---

> ### Author Response · Authors · 2023-11-17
> **Responses to "Clarifying my question about $\beta$-regularity"**
>
> **Our Response**: Thanks for the clarifications. The singular value decomposition (SVD) of general conditional expectation operators is non-trivial compared with its counterpart for matrices. To provide some intuitive explanations, we focus on the exponential family. More specifically, the Gaussian distribution in our response. Suppose we can factorize a joint distribution $p(x,y)$ into $p(x|y)p(y)$, where
>
> $$
> p(x|y)=\frac{1}{\sqrt{2\pi\sigma^2}}\exp(-\frac{(x-y)^2}{2\sigma^2}),\quad p(y)=\frac{1}{\sqrt{2\pi\sigma_0^2}}\exp(-\frac{y^2}{2\sigma_0^2})
> $$
>
> In fact, this joint distribution $p(x,y)$ corresponds to the model $x=y+\epsilon$, where the random variable $y\sim\mathrm{Norm}(0,\sigma_0^2)$ is independent of the random Gaussian noise term $\epsilon\sim\mathrm{Norm}(0,\sigma^2)$. Through SVD of the conditional expectation operator $T$ (associated with $p(x|y)$), **the singular values have a closed form and are given by** (see Makur and Zheng 2016 for more details)
>
> $$
> \lambda_i=\left(\frac{\sigma_0^2}{\sigma_0^2+\sigma^2}\right)^{i/2}.
> $$
>
> We can see that the decay rate of $\lambda_i$ depends on the ratio $\sigma^2/\sigma_0^2$. Recall the model behind the joint distribution is $x=y+\epsilon$, where $\sigma^2$ is the variance of $\epsilon$ and can be seen as noise; and $\sigma_0^2$ is the variance of $Y$ and can be seen as information. Thus, we can see that if the **noise dominates the information** (i.e., $\sigma^2\gg\sigma^2_0$), the decay is fast. On the other hand, if the **information dominates the noise** (i.e., $\sigma_0^2\gg\sigma^2$), the decay is slow.
> If $\beta$ is given, and the noise-to-information ratio is high, the $\beta$-regularity space becomes smaller. In other words, it becomes more difficult to identify functions in the high-noise setting. The reason of the resulting smaller space is that we require the Fourier coefficients to decay at a fast rate to ensure the series is finite.
>
> Now, back to the general model $p(x,y)$, qualitative speaking, the singular values $\lambda_i$ of the operator $T$ describe how well we know about $x$ based on the information of $y$. If $y$ does not provide much information about $x$, or in other words, $x$ contains a lot of noise, the singular values decay fast, and we say the conditional expectation operator is severely ill-posed, and vice versa.
>
> The $\beta$-regularization spaces, as we have explained in the previous responses, are to characterize both the singular values of the conditional expectation operator $T$ and the Fourier coefficients of the function $\rho(\cdot)$. The appropriate value for $\beta$ also depends on both $T$ and $\rho(\cdot)$. A larger $\beta$ implies a smaller function space for $\rho(\cdot)$ to live in, given the operator $T$. Lastly, under our identifiability assumption, we would like to point out that the $\beta$-regularity space can also be viewed as the range of the operator $(T^\ast T)^{\beta/2}$.
>
> ### **Reference**
>
> Makur, A., & Zheng, L. (2016). Polynomial spectral decomposition of conditional expectation operators. In *2016 54th Annual Allerton Conference on Communication, Control, and Computing (Allerton)*. IEEE.

---

> ### Author Response · Authors · 2023-11-17
> **Responses to "Clarifying the "gap" between the  $L^2$ error and test-domain performance"**
>
> Our Response: Thanks for your thought-provoking insights. First, we apologize for the confusion caused by our notation and terminology. Our norm is different, unlike the usual $L^2$ norm of functions, defined as $\Vert f\Vert=\sqrt{\int f(x)^2 dx}$, we define the inner product for the Hilbert space as $\langle f(x,y), g(x, y)\rangle = \int f(x,y) g(x, y) p_s(x,y)dxdy=\mathbb{E}_s(f(x,y)g(x,y))$ and norm as $\Vert f\Vert=\langle f, f\rangle^{1/2}$ (see page 3 of our paper). Thus, the norm we use for a function $f(y)$ is defined as $\Vert f\Vert=\sqrt{\mathbb{E}_s(f(y))^2}$, which is exactly the “weighted $L^2$ norm” you mentioned in your comments.

---

> ### Author Response · Authors · 2023-11-20
> **Thanks for reviewing our paper and providing discussion! Please let us know if there is any further questions for our revised paper and response.**
>
> Dear Reviewer SE3s,
>
> We extend our sincere appreciation for your invaluable feedback and constructive suggestions. As we approach the conclusion of the rebuttal/discussion phase, we would like to inquire if there are any remaining questions or concerns that we can address or clarify.
>
> If you find our revision and response satisfactory, could you please consider raising the score to support our efforts? Your support for our work and insightful suggestions for improving its quality have been immensely valuable to us. We are grateful for the time and consideration you have dedicated to our submission.
>
> Thank you once again.
>
> Authors.

---

> > ### Comment · Reviewer_SE3s · 2023-11-21
> > **Thanks to the authors for their responses.**
> >
> > I think the authors have done a good job responding to the reviewers' comments and revising the paper, so I am raising my rating from 6 to 8. I hope the authors will add some discussion of (a) $\beta$-regularity in the Gaussian example and (b) how exactly estimating $\omega$ in (weighted) $L_2$ relates to test-domain generalization error of the weighted empirical risk minimizer (since, as the authors pointed out, their result is already in weighted $L_2$ norm, I guess this is easy, just following the logic I laid out above).

---

> ### Author Response · Authors · 2023-11-22
> **Thanks for your suggestion and supporting!**
>
> We sincerely thank you for your invaluable feedback and guidance throughout the review process. Your insights have been incredibly helpful in improving our paper, and we have **implemented all the suggestions you provided accordingly in the revision**: We added the discussion on 1) $\beta$-regularity in supplementary Section A.1 and 2) the relationship between the weighted $L^2$ norm and test-domain generalization error in supplementary Section B.1.
>
> Once again, thank you for your time, commitment, and valuable suggestions. We appreciate your dedication to ensuring the highest standards and are grateful for your support.

---

> > ### Comment · Reviewer_SE3s · 2023-11-22
> >
> > Thanks for making the revisions. One small error: in the new Appendix A.1, the phrase "as we have explained in the previous responses," should be removed. :)

---

> > > ### Author Response · Authors · 2023-11-22
> > >
> > > Thanks for pointing out the issue! We have fixed it and updated our supplementary.

---

### Official Review · Reviewer_4YDN · 2023-10-30

**Soundness:** 3 good
**Presentation:** 4 excellent
**Contribution:** 3 good
**Rating:** 8
**Confidence:** 4

**Summary:**

This manuscript studies the continuous target shift problem. To adapt to the target shift, the authors reweighed the training samples with the density ratio of the responses: $\omega(y)=p_t(y)/p_s(y)$. To estimate the weight $\omega(y)$, a two-step procedure is proposed. First, the authors employed a kernel density estimator to estimate the conditional density and then used the outcomes to solve a regularized least-squares problem to obtain the weights. Additionally, the authors discussed conditions for identifiability of the weight $\omega(y)$ and the consistency of the estimation results.

**Strengths:**

Overall, the manuscript is easy to follow and is technically sound. The empirical results also show improvement compared to the baselines.

**Weaknesses:**

I found the major issue is insufficient comparison with other prior works.
For example, Nguyen et al., (2016) studied the same problem with a slightly different estimation procedure. It would be nice to discuss and compare the method.

While the estimation procedure is straightforward, it would be nice to clarify what is the technical challenge and the novelty of this method.




Nguyen, T. D., Christoffel, M., & Sugiyama, M. (2016, February). Continuous target shift adaptation in supervised learning. In Asian Conference on Machine Learning (pp. 285-300). PMLR.

**Questions:**

how to select $\alpha$ in terms of the sample size?

---

> ### Author Response · Authors · 2023-11-16
> **Responses to Reviewer 4YDN**
>
> ### Comment 1
>
> > *I found the major issue is insufficient comparison with other prior works. For example, Nguyen et al., (2016) studied the same problem with a slightly different estimation procedure. It would be nice to discuss and compare the method.*
> >
> >
> > *While the estimation procedure is straightforward, it would be nice to clarify what is the technical challenge and the novelty of this method.*
> >
> > *Nguyen, T. D., Christoffel, M., & Sugiyama, M. (2016, February). Continuous target shift adaptation in supervised learning. In Asian Conference on Machine Learning (pp. 285-300). PMLR.*
> >
>
> ### Our Response
>
> In our revised paper, we **added** **the** **L2IWE method** proposed by Nguyen et al. (2016) for extensive comparisons. Our proposed ReTaSA method still **outperforms** the L2IWE method consistently. The KMM and the L2IWE methods are similar in spirit as they both try to estimate the importance weight function through distribution matching; their difference is that they choose different function spaces to search for the importance weight function. The **novelty** of our method is that instead of matching, we formulate the problem as a task of solving an equation. Also, our proposed method only involves a one-step matrix inversion rather than an iterative optimization procedure.
> The curse of dimensionality may pose some **challenges**. However, this problem can be addressed by adopting the black-box-shift estimation method, as detailed in the supplementary.
>
> ### Comment 2 (Questions)
>
> > How to select $\alpha$ in terms of the sample size?
> >
>
> ### Our Response
>
> Thanks for pointing this issue out. We agree that hyperparameter tuning is not trivial. However, based on our Theorem 1, the regularization parameter needs to satisfy the condition that $\lim_{n\to\infty}n\alpha^2=\infty$. Thus, in our experiment, we proposed to choose $\alpha=n^{-1/4}$ as a rule of thumb to meet the condition for practical simplicity. In Section C.2.3 of the supplementary materials, we conducted a sensitivity analysis from which $\alpha=n^{-1/4}$ achieves good performance. We use this rule of thumb for all the numerical studies, and the results are consistently good.

---

> ### Author Response · Authors · 2023-11-20
> **Thanks for reviewing our paper! Any further questions for our revised paper and response?**
>
> Dear Reviewer 4YDN,
>
> We extend our gratitude for your insightful feedback on our work. Your comments have been invaluable in shaping the refinement of our paper. Here's a summary of the key modifications we made in response to your suggestions:
>
> 1. Inclusive comparison: We introduced the L2IWE method proposed by Nguyen et al. (2016) for a comprehensive comparative analysis.
> 2. Enhanced clarity on novelty and challenge: We redefined the non-trivial target shift problem as a solvable linear equation problem, supported by a robust theoretical foundation.
> 3. Efficient hyper-parameter tuning: We introduced a simple yet effective rule of thumb method for selecting the regularization parameter, along with a thorough ablation study.
>
> With the end of the rebuttal phase approaching, we kindly invite you to review our responses and the revised paper. Your feedback is crucial to ensuring that our clarifications address the concerns you raised. If our revision and response meet your satisfaction, would you consider raising the score to support our work? We highly value your support and guidance in elevating the quality of our work.
>
> Thank you for your time and consideration.
>
> Authors.

---

> ### Comment · Reviewer_4YDN · 2023-11-22
>
> Thanks for addressing my comments and implementing the new baseline. The proposed method outperforms the baselines.
>  If possible, it would be useful to discuss why L2IWE fails to beat non-adaptation in the SOCR and Szeged Weather dataset as Table 1 shows negative $\Delta Pred$ MSE. Does any numerical instability occur in the estimation procedure? Then, it would be nice to address why the proposed method does not fail. As other reviewers pointed out, the kernel ratio density estimator could be unstable sometimes. More discussions on numerical stability can be helpful for practitioners to choose between algorithms.
>
> Overall, I think the paper is well-written and the contribution is clear, so I raise my score a bit.

---

> ### Author Response · Authors · 2023-11-23
> **Thank you!**
>
> We appreciate your support for our work! Regarding the performance of L2IWE on the SOCR and Szeged Weather datasets, we observed that L2IWE's performance will decrease drastically when the shift becomes more severe. In Section C.3 of our supplementary materials, we presented several examples illustrating the estimated weights for each method. Notably, L2IWE exhibited higher weights concentrated in the middle of the response (i.e., $y$) support, with near-zero weights at the boundaries. However, in these experiments, the oracle weights shifted towards the boundaries. In summary, empirical evidence shows that L2IWE's performance is unsatisfactory under severe shifts. The shifts in the real datasets are more severe; we conjecture that this may contribute to the underperformance of the L2IWE method. Due to the time limit of the discussion period, we do not have enough time to find the causes more in-depth, but we shall investigate this later.
>
> Additionally, we acknowledge the potential impact of ratio density estimation on our algorithm's performance. In Section C.2.3 of the supplementary materials, we conducted a sensitivity study involving two density ratio estimation methods. While we did not observe significant performance changes, we recognize the importance of a more in-depth analysis and discussion on numerical stability. Further exploration in this aspect would contribute to a comprehensive understanding of our algorithm's behavior.
>
> Thanks again for your time and efforts!

---

### Official Review · Reviewer_2aLy · 2023-10-30

**Soundness:** 3 good
**Presentation:** 3 good
**Contribution:** 3 good
**Rating:** 6
**Confidence:** 3

**Summary:**

This paper studies the problem of label shift within a continuous label space, such as in regression with label shift. To tackle this issue, the authors propose a method based on importance weighting, which transforms the learning task into an estimation problem concerning the density ratio of the continuous variable $y$. The method has statistical consistency in estimating the weight function under certain identifiability assumptions. A practical algorithm is derived from the theoretical analysis.

**Strengths:**

1. This paper considers an interesting and important real-world problem with a compelling motivation.
2. This work provides the statistical consistency analysis for the proposed estimator in the continuous label space. This technical analysis has the potential insights to benefit studies in this area.

**Weaknesses:**

1. In the experiments, only low-dimensional and small-scale datasets are used for comparison. Thus, the empirical results are not sufficient to support the success of the proposed algorithm. Considering the increasing need of learning with high-dimensional data such as images and the poor performance of kernel density approximation in high-dimensional data, the comparison in high-dimensional data is important.

2. Determining the hyperparameter $\alpha$ for different data sets is difficult.

3. In classical ratio estimation approaches, the estimation variance is sometimes large, e.g. $p_s(y)$ goes to zero, making the algorithm unstable. In the continuous label space, it is unclear whether this problem becomes worse. It is suggested to discuss this issue or to introduce an additional variance reduction mechanism.

**Questions:**

See the comments in weakness part. As also, it is suggested to test the performance of the proposed algorithm on large-scale and high-dimensional datasets.

---

> ### Author Response · Authors · 2023-11-16
> **Responses to Reviewer 2aLy**
>
> ### Comment 1
>
> > *In the experiments, only low-dimensional and small-scale datasets are used for comparison. Thus, the empirical results are not sufficient to support the success of the proposed algorithm. Considering the increasing need of learning with high-dimensional data such as images and the poor performance of kernel density approximation in high-dimensional data, the comparison in high-dimensional data is important.*
> >
>
> ### Our Response
>
> In the revision, we have demonstrated the effectiveness of our proposed method with larger datasets with higher dimensions (e.g., high dimension regression problem on UCI community and crime dataset and knowledge distillation regression problem on MNIST dataset). Due to space limits, we put the **additional numerical studies** in **Sections C.4.1 & 4.2** of the supplementary materials. Furthermore, to handle the density approximation problem with high-dimensional data, we adopted the black-box-shift-estimation method Lipton et al. (2018) proposed in our setting. Please refer to **Section C.4** for the details.
>
> ### Comment 2
>
> > *Determining the hyperparameter $\alpha$ for different data sets is difficult.*
> >
>
> ### Our Response
>
> Thanks for pointing this issue out. Based on Theorem 1, the regularization parameter must satisfy the condition that $\lim_{n\to\infty}n\alpha^2=\infty$. Thus, in our experiment, we proposed to choose $\alpha=n^{-1/4}$ as a rule of thumb to meet the condition for practical simplicity. In Section C.2.3 of the supplementary materials, we conducted a sensitivity analysis from which $\alpha=n^{-1/4}$ achieves good performance. We use this rule of thumb for all the numerical studies, and the results are good.
>
> ### Comment 3
>
> > *In classical ratio estimation approaches, the estimation variance is sometimes large, e.g. $p_s(y)$ goes to zero, making the algorithm unstable. In the continuous label space, it is unclear whether this problem becomes worse. It is suggested to discuss this issue or to introduce an additional variance reduction mechanism.*
> >
>
> ### Our Response
>
> Indeed, it is problematic when $p_s(y)$ goes to zero. This is why we need the assumption that $p_s(y)$ is bounded from below by some constant $\epsilon>0$ (see Assumption 3). Also, we need the support of $p_t(y)$ to be a subset of the support of $p_s(y)$. When $p_s(y)$ goes to zero (while $p_t(y)$ is not zero), it corresponds to the out-of-distribution (OOD) setting. Although this is beyond the scope of our paper, it represents an interesting direction for future research. We discussed this issue as a future research direction in the final discussion.
>
> ### Comment 4 (Questions)
>
> > *See the comments in weakness part. As also, it is suggested to test the performance of the proposed algorithm on large-scale and high-dimensional datasets.*
> >
>
> In this revision, we **applied our method to larger datasets with higher dimensions** in additional experiments. More specifically, we include the experiments of high dimension regression problem on the UCI community and crime dataset and knowledge distillation regression problem on the MNIST dataset in the supplementary. The proposed method demonstrated good performance. Due to space limits, the numerical study results can be found in **Sections C.4.1 & 4.2** of the supplementary materials.

---

> ### Author Response · Authors · 2023-11-20
> **Thanks for reviewing our paper! Any further questions for our revised paper and response?**
>
> Dear Reviewer 2aLy,
>
> We express our gratitude for your time and efforts for reviewing our paper and offering valuable suggestions. In our rebuttal, we delved deeper into three key areas:
>
> 1. Empirical performance comparison on large-scale and high-dimensional datasets.
> 2. Elaboration on the regularization parameter selection, along with its ablation study.
> 3. Further extension of discussions regarding $p_s(y)\rightarrow 0$ and addressing out-of-distribution (OOD) problems.
>
> As we approach the conclusion of the discussion phase, we hope you have had the opportunity to read our responses and the revised paper. In consideration of time constraints, we hope you can inform us of any lingering questions or concerns that require additional attention or clarification. If our revision and response meet your satisfaction, we would greatly appreciate your consideration in raising the score.
>
> Your time and consideration are highly appreciated.
>
> Authors.

---

### Official Review · Reviewer_D6fP · 2023-11-01

**Soundness:** 3 good
**Presentation:** 4 excellent
**Contribution:** 3 good
**Rating:** 6
**Confidence:** 3

**Summary:**

This work addresses the challenge of distribution shifts in deploying modern machine learning models, specifically focusing on the target shift problem within a regression context. It tackles the situation where the continuous target variable y exhibits different marginal distributions between the training and testing domains, while the conditional distribution of features x given y remains constant. Notably, the regression problem's infinite-dimensional target space necessitates unique solutions. The authors propose ReTaSA, a nonparametric regularized approach to estimate the importance weight function and provide theoretical justification for it, thereby effectively addressing the continuous target shift problem. Extensive numerical studies on both synthetic and real-world datasets confirm the effectiveness of the proposed method.

**Strengths:**

- The work focuses on addressing the challenge of classification tasks with an infinite-dimensional target space, while previous research on target shift primarily concentrated on scenarios where $y$ is categorical. This problem holds fundamental significance across various domains.

- By precisely estimating the importance weight function, the model can effectively adapt to variations in the target variable's distribution between the training and testing domains. A continuous importance weight function offers greater flexibility compared to discrete methods.

- The authors provide thorough theoretical justifications for consistency and error rate bounds, enhancing the paper's rigor and reliability.

- The paper's organization is well-structured, maintaining a clear and easily-followed flow from start to finish.

- Extensive references to related work offer a comprehensive overview of prior research, providing valuable context for the study and highlighting the authors' deep understanding of the field.

**Weaknesses:**

- Given the importance weight function, is the estimation process sensitive to noise or outliers in the data and thus potentially leading to inaccurate estimates of the importance weight function? Or this limitation is conquered by applying the regularization technique in your paper?

 - Estimating a continuous importance weight function is a probabilistic process, together with the regularization process, this may inherent uncertainty in the estimates, affecting the model's reliability in certain cases. How do you solve this issue?

 - Regularization can help stabilize the estimation process, but accurate estimation of a continuous importance weight function may still require a substantial amount of data from both source and target domains. What is the sample complexity of your algorithm?

**Questions:**

See above "weaknesses" section.

---

> ### Author Response · Authors · 2023-11-16
> **Responses to Reviewer D6fP**
>
> ### Comment 1
>
> > *Given the importance weight function, is the estimation process sensitive to noise or outliers in the data and thus potentially leading to inaccurate estimates of the importance weight function? Or this limitation is conquered by applying the regularization technique in your paper?*
> >
>
> ### Our Response
>
> Yes, the noise and outlier data will affect the estimation performance and the regularization applied in our method can enhance our ability to robustly adjust to uncertainties in estimating the importance weight function.
>
> Previous studies have illustrated such advantage of regularization in classification tasks; for instance, Azizzadenesheli et al. (2019) introduced a regularized version of the label shift estimator called RLLS, building upon the BBSE method of Lipton et al. (2018), who did not employ regularization. The regularized estimator RLLS demonstrated superior performance to the original non-regularized version BBSE, indicating that the regularization does help.
>
> In the continuous continuous, regularization serves a **dual** purpose: addressing the ill-posedness problem while providing the advantage of robustness. However, unlike the classification case where regularization is optional (i.e., Lipton et al. 2018), the problem we consider becomes ill-posed due to the 'explosive behavior' in the continuous case without regularization, as detailed in our paper. So, regularization is a must-have in our case.
>
> ### Comment 2
>
> > *Estimating a continuous importance weight function is a probabilistic process, together with the regularization process, this may inherent uncertainty in the estimates, affecting the model's reliability in certain cases. How do you solve this issue?*
> >
>
> ### Our Response
>
> The functional estimation, and indeed any estimator, is inherently contingent upon the randomness of the data sample; uncertainty becomes an inherent characteristic of the estimation. In our study, **we establish a theoretical bound for the estimator’s convergence rate in Theorem 1**, and the error bound measures the estimation uncertainty from the theoretical aspect. **Also, in our numerical evaluation, we presented all the estimation performance with confidence intervals and standard deviations, which can capture the estimation uncertainty.**
>
> A related aspect involves the uncertainty quantification for the continuous case. For instance, considering the randomness inherent in the estimation, how to construct a prediction set for the response. While this lies beyond the scope of our current paper, it represents an interesting direction for future research. Existing efforts have been made on uncertainty quantification in the classification case (e.g., Podkopaev and Ramdas 2021), but notably, there appears to be a gap in addressing the continuous case. We have added a discussion on future research in the paper.
>
> ### Comment 3
>
> > *Regularization can help stabilize the estimation process, but accurate estimation of a continuous importance weight function may still require a substantial amount of data from both source and target domains. What is the sample complexity of your algorithm?*
> >
>
> ### Our Response
>
> We agree that the sample sizes play an important role in the weight estimation. Theorem 1 provides a stochastic order bound for our proposed functional estimation, in which the sample size will directly affect the convergence rate. In Equation (11), the bound is a function of $n$ (sample size of the source data) and $m$ (sample size of the target data) given other quantities (e.g., the tuning parameter $\alpha$ and the bandwidth $h$).
>
> ### References
>
> Lipton, Z., Wang, Y. & Smola, A.. (2018). Detecting and Correcting for Label Shift with Black Box Predictors. *ICML*.
>
> Azizzadenesheli, K., Liu, A., Yang, F., & Anandkumar, A. (2019). Regularized learning for domain adaptation under label shifts. *ICLR*.
>
> Podkopaev, A., & Ramdas, A. (2021). Distribution-free uncertainty quantification for classification under label shift. *UAI*.

---

> ### Author Response · Authors · 2023-11-20
> **Thanks for reviewing our paper! Any further questions for our revised paper and response?**
>
> Dear Reviewer D6fP,
>
> We would like to thank you for your thorough review and insightful suggestions. In response to your feedback, we have incorporated additional discussion points in our rebuttal:
>
> 1. Explanation of how regularization contributes to estimation robustness.
> 2. Detailed insights into the measurement and quantification of estimation uncertainty.
> 3. Clarification of the theoretical sample complexity of our proposed method.
>
> As we near the conclusion of the rebuttal phase, we kindly invite you to take a moment to review our responses. We are eager to know if our efforts have addressed and clarified the concerns you raised. If our revision and response align with your satisfaction, we would sincerely appreciate your consideration in raising the score.
>
> Your time and consideration are greatly appreciated.
>
> Thank you,
>
> Authors.

---

### Author Response · Authors · 2023-11-16
**Summary of the Revision**

We would like to express our gratitude to the reviewers for their invaluable feedback and insightful comments on our paper “**ReTaSA: A Nonparametric Functional Estimation Approach for Addressing Continuous Target Shift**”. Their thoughtful assessment has significantly contributed to enhancing the quality of our work.

We first would like to thank the reviewers for acknowledging the **strengths** of our paper.

- Addresses classification tasks with an infinite-dimensional target space, a fundamental challenge in various domains often overlooked in prior research. [Reviewers D6fP, 2aLy, SE3s]
- Provides robust theoretical justifications for consistency and error rate bounds, enhancing the paper's reliability. [Reviewers D6fP, 2aLy, 4YDN, SE3s]
- Maintains a well-structured organization, ensuring a clear and easily-followed flow throughout the paper. [Reviewers SE3s, 4YDN, D6fP]
- Offers an extensive reference base, providing a comprehensive overview of related work, demonstrating a profound understanding of the field. [Reviewers D6fP, 4YDN]

In the **revision**, we have improved the paper by doing the followings.

- Added another method called L2IWE proposed by Nguyen et al., (2016). Our numerical study shows that our proposed method still has the best performance. [Reviewer 4YDN]
- We applied the proposed method to larger datasets with higher dimension. The ReTaSA method is proven to be effective for high-dimensional data. We address the curse of dimensionality by adapting the black-box-shift-estimation method in Lipton et al. (2018). The results are given in Section C of the supplementary materials. [Reviewers SE3s, 2aLy]
- We have fixed some typos and inconsistent issues of our theoretical results. [Reviewer SE3s]
- We added more discussions on future research directions. [Reviewer SE3s, D6fP]

---

### Meta-Review · Area_Chair_64Rd · 2023-12-06

**Metareview:**

The paper studies the problem of domain adaptation for regression in the context of continuous target shift. They propose a method based on an importance sampling strategy. The weight estimation is based on 2-step process leveraging a kernel density estimator combined with a regularized least-squares optimization problem. A theoretical analysis is provided with a consistency result.

On the positive side, the reviews identified that the idea of addressing of target shift with respect to continuous regression is interesting and relevant for real-world tasks, that the method is sound and clear, that the work is supported by a nice theoretical justification (statistical consistency of the estimator), and the paper is well structured and easy to follow with an experimental evaluation that supports the claims.
On the negative side, there is a lack of precision/discussion on the robustness, the complexity of the method, the assessment of hyper-parameters or difficulty of variance estimation, the theoretical computational complexity; the experimental evaluation is limited to small dimensional datasets and more baselines can be considered, general discussion on the novelty of the method and limitations of the experiments.

Authors did a strong effort in their rebuttal and provided a revision based on reviewers’ comments. The answers provided were positively evaluated. Authors have provided additional experiments (which was an important point)  and detailed answers to the identified weaknesses.
The new revised paper presents a clear and well-written contribution supported by reviewers.

I propose then acceptance.

**Justification For Why Not Higher Score:**

There were many weaknesses identified in the initial submission, authors have provided clear answers, but maybe this makes the paper below than other contributions.
But I'm ok if the paper is accepted as spotlight.

**Justification For Why Not Lower Score:**

Clear positive consensus with reviewers with a score of 7 (minimum score 7)

---

### Decision · Program_Chairs · 2024-01-16

Accept (poster)